# Motion impact score for detecting spurious brain-behavior associations

Benjamin P. Kay [1] ✉, David F. Montez [1], Scott Marek [2,3], Brenden Tervo-Clemmens [4,5,6], Joshua S. Siegel [2,7], Babatunde Adeyemo [1,2], Timothy O. Laumann [2,3], Athanasia Metoki [1], Roselyne J. Chauvin [1], Andrew N. Van [1,8], Vahdeta Suljic [1], Samuel R. Krimmel [1], Ryland L. Miller [1], Dillan J. Newbold [1,9], Annie Zheng [1], Nicole A. Seider [1], Kristen M. Scheidter [1], Julia S. Monk [1], Eric Feczko [4,10], Anita Randolph [4,10], Óscar Miranda-Domínguez [4,10], Lucille A. Moore [4], Anders J. Perrone [4], Gregory M. Conan [4], Eric A. Earl [11], Stephen M. Malone [12], Michaela Cordova [13], Olivia Doyle [14], Benjamin J. Lynch [15], James C. Wilgenbusch [15], Thomas Pengo [16], Alice M. Graham [14], Jarod L. Roland [17], Evan M. Gordon [3], Abraham Z. Snyder [1,3], Deanna M. Barch [2,3,18], Damien A. Fair [4,10,19] & Nico U. F. Dosenbach [1,3,8,18,20]

In-scanner head motion introduces systematic bias to resting-state fMRI functional connectivity (FC) not completely removed by denoising algorithms. Researchers studying traits associated with motion (e.g. psychiatric disorders) need to know if their trait-FC relationships are impacted by residual motion to avoid reporting false positive results. We devised Split Half Analysis of Motion Associated Networks (SHAMAN) to assign a motion impact score to specific trait-FC relationships. SHAMAN distinguishes between motion causing overestimation or underestimation of trait-FC effects. We assessed 45 traits from $n = 7270$ participants in the Adolescent Brain Cognitive Development (ABCD) Study. After standard denoising with ABCD-BIDS and without motion censoring, 42% (19/45) of traits had significant ($p < 0.05$) motion overestimation scores and 38% (17/45) had significant underestimation scores. Censoring at framewise displacement (FD) < 0.2 mm reduced significant overestimation to 2% (1/45) of traits but did not decrease the number of traits with significant motion underestimation scores.

Head motion is the largest source of artifact in structural and functional MRI (fMRI) signals[1–20]. The technical challenge posed by motion cannot be overstated and has motivated the creation of behavioral interventions[9,21–24] and real-time motion tracking software to reduce the amount of in-scanner head motion[5,25]. Even with highly compliant participants, involuntary sub-millimeter head movements systematically alter fMRI data[6,14,17,19,20]. Unfortunately, non-linear characteristics of MRI physics make removal of motion artifact during post-processing difficult[26]. Compared to task fMRI, resting-state functional connectivity (FC) is especially vulnerable to motion artifact because the timing of the underlying neural processes is unknown[2,4,6,9,10,12,14,19,20,27–30]. The effect of motion on FC has been shown to be spatially systematic, causing decreased long-distance connectivity and increased short-range connectivity, most notably in the default mode network[19,20,27,28].

Results from early studies of children, older adults, and patients with neurological or psychiatric disorders have been spuriously related to motion[6,19,29–31]. For example, motion artifact systematically

decreases FC between distant brain regions[28] leading some investigators to conclude that autism decreases long-distance FC when, in fact, their results were due to increased head motion in autistic study participants. These cautionary findings have motivated the creation of numerous approaches to mitigate motion artifact including global signal regression[28,32], motion parameter regression[29], spectral filtering, respiratory filtering[25,33,34], principal component analysis[1,11], independent component analysis[7,15,35], multi-echo pulse sequences[8], despiking of high-motion frames[13], and combinations thereof[3,4,16,25]. However, given the complexity of these approaches, it is difficult to be certain that enough motion artifact has been removed to avoid over- or underestimating trait-FC effects.

It is increasingly common for brain-wide association studies (BWAS) involving many thousands of participants (e.g. HCP, ABCD, UK Biobank) to provide data that have already been processed to remove motion[36–39]; however, even in these cases there is some choice as to how much data to retain or to censor. In such large studies, obtaining the raw data and re-applying a different motion processing method is computationally expensive. Excluding high-motion fMRI frames (timepoints) from analysis by censoring is a post-hoc approach shown to further reduce residual motion artifact[14,20,27,29]. Power et al.[40] and Pham et al.[41] note a natural tension between the need to remove some motion-contaminated volumes to reduce spurious findings (false positive inference) but not so many volumes as to bias the sample distribution of a trait by systematically excluding individuals with high motion who may exhibit important variance in the trait of interest (e.g., low scores on attention measures associated with greater motion).

This difficulty in censoring threshold selection arises in part because most approaches for quantifying motion are agnostic to the hypothesis under study[6,14,19,20,29]. However, some traits or groups of participants are more strongly correlated with motion than others. For example, study participants with attention-deficit hyperactivity disorder or autism have higher in-scanner head motion than neurotypical participants[5,6,9,30,42]. Even when much of the overall signal variance associated with motion has been removed, inferences about such motion-correlated traits may still be significantly impacted by motion artifact[6,9,18,20,29,30]. Therefore, in addition to standard approaches for quantifying motion, methods for quantifying trait-specific motion artifact in FC are needed.

Approaches for quantifying the association of specific trait-FC effects with motion include measuring changes in distance-dependent correlations at different levels of motion censoring[27,28], measuring spatial (i.e., across edges) similarity of trait-FC effects with motion-FC effects[29,30], and measuring differences in trait-FC effects between groups matched according to levels of motion[6,19]. However, these measures do not establish a threshold for acceptable or unacceptable levels of trait-specific motion. Nielsen et al.[12] proposed using support vector machines to test whether head motion is significantly predicted by FC, but the method applies only to high-order multivariate models of trait-FC effects. Siegel et al.[18] proposed conceptualizing the relationship between motion and trait-FC effects by comparing within-participant and between-participant variance in the trait-FC effects explained by motion. Siegel's original method was limited because it required repeated resting-state fMRI (rs-fMRI) scans of the same participant on different days, and it used a simple correlation measure that could not account for covariates or distinguish between motion artifact causing over- or under-estimation of the trait-FC effects.

Thus, we developed a novel method for computing a trait-specific motion impact score that operates on one or more rs-fMRI scans per participant and can be adapted to model covariates. We capitalize on Siegel et al.'s[18] observation that traits (e.g. weight, intelligence) are stable over the timescale of an MRI scan whereas motion is a state that varies from second to second. The proposed **S**plit **H**alf **A**nalysis of **M**otion **A**ssociated **N**etworks (SHAMAN) capitalizes on the relative stability of traits over time by measuring difference in the correlation structure between split high- and low-motion halves of each participant's fMRI timeseries. When the trait-FC effects are independent of motion, the difference in each half of the connectivity will be not-significant because traits are stable over time. A significant difference is detected only when state-dependent differences in motion have an impact on the trait's connectivity. A direction (positive or negative) of the motion impact score that is aligned with the direction of the trait-FC effect is consistent with motion causing overestimation of the trait-FC effect, a "motion overestimation score." A motion impact score opposite the direction of the trait-FC effect is consistent with motion causing underestimation of the trait-FC effect, a "motion underestimation score." Permutation of the timeseries and non-parametric combining[43,44] across pairwise connections yields a motion impact score and a p-value distinguishing significant from not-significant impacts of motion on trait-FC effects.

Recently the Adolescent Brain Cognitive Development (ABCD) Study collected up to 20 minutes of rs-fMRI data on 11,874 children ages 9–10 years[21,45,46] with extensive demographic, biophysical, and behavioral data[47,48]. Such large data sets have made it possible to quantify reproducibility in resting-state fMRI, and they have revealed that the true effect sizes of brain-wide association studies (BWAS) are smaller than previously thought due to sampling variability[49]. Failure to consider head motion is another source of inconsistent results. Thus, we first characterized the effectiveness of standard denoising approaches at reducing motion artifact. Then we considered the residual trait-specific impact of head motion on FC in the high-quality ABCD data after denoising and varying levels of post-hoc motion censoring. We performed supplementary analyses on the Human Connectome Project[50–52] to demonstrate the generalizability of our results to other denoising methods and data sets.

## Results

### The Effect of Residual Motion is Large Even After Denoising and Censoring

In order to characterize the impact of residual head motion on trait-FC effects, we first performed preliminary analyses to quantify how much residual motion was left in the data after denoising. Of the 11,874 children recruited into ABCD, n = 9652 children with at least 8 minutes of rs-fMRI data were included in this portion of the analysis. ABCD-BIDS is the default denoising algorithm for pre-processed ABCD data[25,37]. It includes global signal regression, respiratory filtering, spectral (low-pass) filtering, despiking, and regressing out the motion parameter timeseries. The relative performance of ABCD-BIDS was evaluated by comparing how much of the between-participant variability in the fMRI timeseries (averaged across regions of interest) was explained by head motion (framewise displacement, FD) in a linear, log-log transformed model before and after applying ABCD-BIDS (Supplementary Fig. 1). After minimal processing[39] (i.e. motion-correction by frame realignment only), 73% of signal variance (as estimated by taking the square of Spearman's rho) was explained by head motion. After further denoising using ABCD-BIDS (respiratory filtering, motion timeseries regression, despiking/interpolation of high-motion frames), 23% of signal variance was explained by head motion. Therefore, ABCD-BIDS achieved a relative reduction in the proportion of signal variance related to motion of 69% compared to minimal processing alone (see Methods).

However, even after denoising with ABCD-BIDS, it was still possible to detect large between-participant differences associated with head motion. The average FC matrix across all participants is shown in Fig. 1a, b. The residual motion-FC effect was quantified by regressing each participant's FD (averaged over all resting-state scans) against their FC to generate a motion-FC effect matrix with units of change in FC per mm FD (Fig. 1c, d). The average FC matrix and motion-FC effect matrix were compared by computing the correlation between their edges (pairwise connections). The motion-FC effect matrix had a

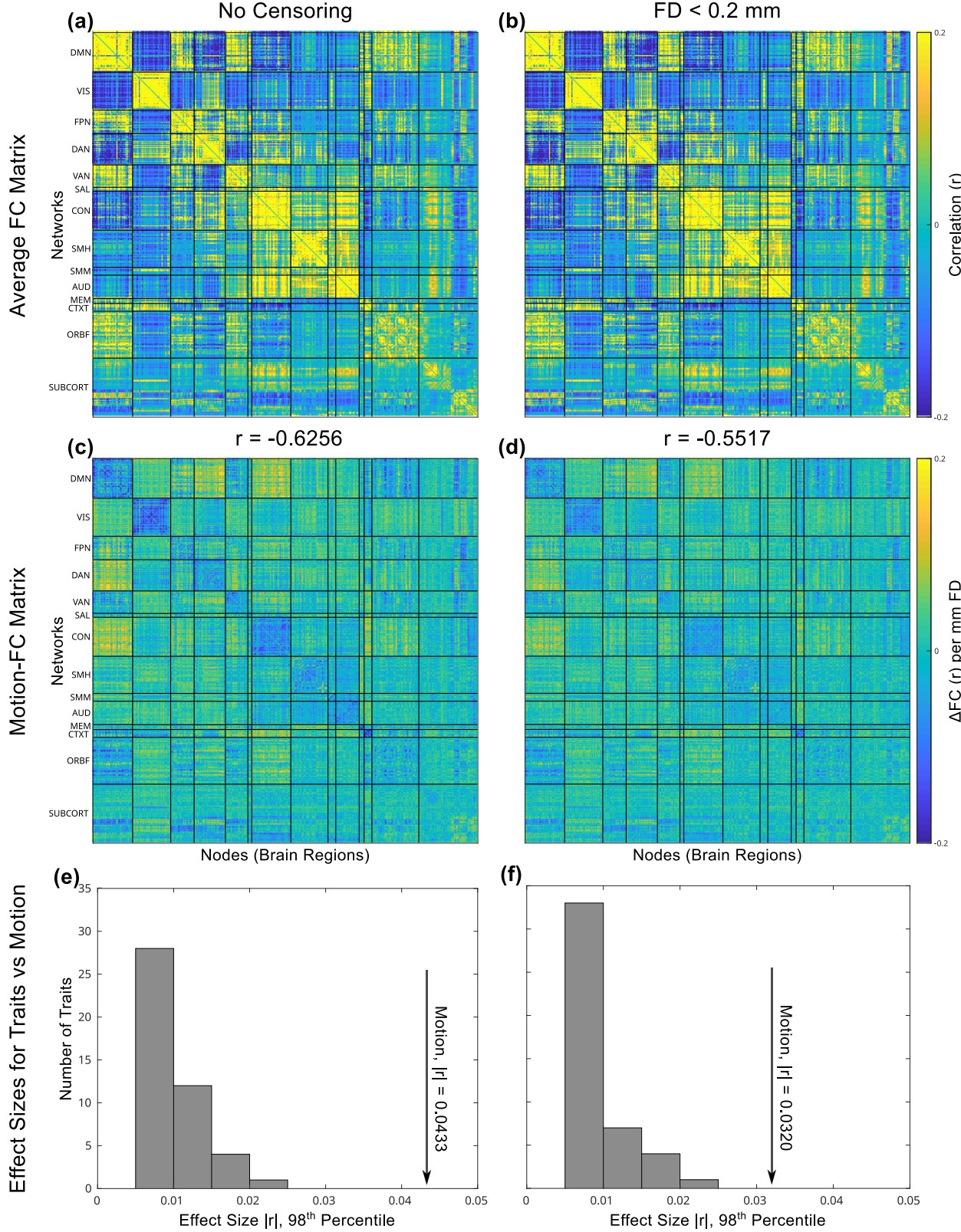

strong, negative correlation of Spearman ρ = −0.58 with the average FC matrix. This strong, negative correlation persisted even after motion censoring at FD < 0.2 mm (Spearman ρ = −0.51). Thus, across all functional connections, connection strength tended to be weaker in participants who moved more compared to participants who moved less.

The decrease in FC due to head motion was larger than the increase or decrease in FC related to traits of interest (Fig. 1e, f). The

largest motion-FC effect size for a single connection was |r| = 0.10. More conservatively, the 98th percentile for the motion-FC effect size was |r| = 0.04. While these effect sizes were small in absolute terms, Marek et al.[49] have shown that the largest 1% of univariate trait-FC effect sizes in ABCD are on the order of |r| = 0.06. Of the 45 variables we evaluated, parental income bracket had the largest trait-FC effect size at |r| = 0.06 (98th percentile |r| = 0.02). With motion censoring at

**Fig. 1 | Average connectivity matrix, motion-FC effect matrix, and effect sizes.** **a** The average (across n = 7270 participants) correlation matrix for functional connectivity (FC), after denoising with ABCD-BIDS without frame censoring, see Supplementary Fig. 10 to visualize average FC matrices from high- and low-motion halves of the data separately; **b** and with n = 6,886 participants after frame censoring at framewise displacement (FD) < 0.2 mm. **c** The motion-FC effect matrix obtained by regressing average FD against FC for each participant, FC ~ 1 + FD. The motion-FC effect matrix has units of change in FC per mm FD. **d** Motion-FC effect matrix with censoring at FD < 0.2. **a**, **c** The upper triangular parts of the FC connectivity matrices were edge-for-edge correlated at Pearson r = −0.6256 (Spearman ρ = −0.5822). **b**, **d** After motion censoring at FD < 0.2 mm, the average FC and

motion-FC effect matrices were edge-for-edge correlated at Pearson r = −0.5517 (Spearman ρ = −0.5059). **e** The effect size of the motion-FC effect (arrow) is plotted relative to the empirical distribution of effect sizes for the 45 traits in this study without motion censoring and **f** with motion censoring at framewise displacement (FD) < 0.2 mm. Effect sizes were computed as the largest (98th percentile) normalized difference in functional connectivity (correlation, |r|) across edges for each trait. Networks: DMN Default Mode Network, VIS Visual, FPN Frontoparietal Network, DAN Dorsal Attention Network, VAN Ventral Attention Network, SAL Salience, CON Cingulo-Opercular Network, SMH Somatomotor Hand, SMM Somatomotor Mouth, AUD Auditory, MEM Parietal Memory Network, CTXT Context, ORBF Orbitofrontal, SUBCORT subcortical regions.

FD < 0.2 mm the largest motion-FC effect was |r| = 0.08 (98th percentile |r| = 0.03) and the largest effect size for parental income was |r| = 0.06 (98th percentile |r| = 0.02). Therefore, even with motion denoising and censoring, the motion-FC effect was large in relation to trait-FC effects.

## Most Traits Were Significantly Correlated with Head Motion

Of the 11,874 total children recruited into ABCD, and 9,652 children with at least 8 minutes of rs-fMRI data, n = 7270 children without missing data for any of the 45 traits we examined were included in this and subsequent analyses.

Surprisingly, many seemingly unrelated traits were shown to be correlated with head motion in the Human Connectome Project (HCP)[18]. In ABCD, 87% (39/45) of traits or variables evaluated showed significant (p < 0.05) correlations between the trait (e.g. BMI) and head motion (FD, in mm; Supplementary Data 1). WISC-V (Wechsler Intelligence Scale for Children, 5th edition) matrix reasoning subscore[53] had the highest trait-FD correlation at Pearson r = −0.12, Spearman ρ = −0.12, p < 0.001, and Child Behavioral Checklist (CBCL) somatization subscore had the least trait-FD correlation at Pearson r = 0.0004, Spearman ρ = −0.02 (not significant).

Body mass index (BMI)[54] and matrix reasoning subscore[53] were selected as exemplar traits. Both exhibited significant trait-FD correlations (Supplementary Fig. 2). Children with higher BMI exhibited more in-scanner head motion (Spearman ρ = 0.13, p < 0.001). Children with higher matrix reasoning scores had lower head motion (Spearman ρ = −0.12, p < 0.001).

## SHAMAN Detected the Impact of Motion on Trait-FC Effects in Simulated Data

To clarify and validate the principle of the SHAMAN approach, a generative model was used to simulate fMRI timeseries data with experimentally-controlled amounts of "BRAIN" trait-FC effects, "MOTION" motion-FC effects, and relationship between them. Simulation of non-linear relationships between the trait-FC effects and motion caused denoising to fail and for SHAMAN to detect a significant impact of residual motion on trait-FC effects (motion impact score) as depicted in Fig. 2. The SHAMAN algorithm is described further in the Methods, and simulation methods are described further in the Supplementary Material. Briefly, the FC matrix for a simulated participant contained both "BRAIN" and "MOTION" effects (Fig. 2a), reflecting the failure of denoising to completely remove the "MOTION" effects. The participant's fMRI timeseries was split into high- and low-motion halves based on the FD timeseries. More "MOTION" was visible in the FC matrix from the high-motion half of the timeseries compared to the low-motion half, whereas the amount of "BRAIN" did not vary over time and was therefore was equal between halves (Fig. 2b). When the low-motion trait-FC matrix was subtracted from the high-motion trait-FC effect matrix, the identical "BRAIN" trait-FC effects canceled out in the difference FC matrix (Fig. 2d) whereas the "MOTION" did not. Finally, the "BRAIN" trait was regressed against the difference FC matrices to generate a matrix of motion impact score (Fig. 2e). The motion impact score approximates how estimates of the trait-FC effects are changed by head motion (Supplementary Fig. 16d,f). A

motion overestimation score approximates how much the trait-FC effect is overestimated due to motion. A motion underestimation score approximates how much the trait-FC effect is underestimated due to motion.

## Residual Motion Inflates Functional Connectivity Associations with Biophysical Traits

We found that residual motion-related signal inflated the effect size of many traits, especially the biophysical traits of BMI, age, sex, weight, and height. Trait-FC effects were calculated separately for each of these traits on fMRI data after denoising with ABCD-BIDS using conventional methods with mean framewise displacement (FD) as a covariate to control for between-participant differences in motion (e.g., FC ~ BMI + FD). Without frame censoring, all five biophysical traits had significant motion overestimation scores, meaning that their trait-FC effects were significantly (p < 0.05) greater due to the impact of residual head motion. Three biophysical traits, BMI, sex, and weight, also had significant motion underestimation scores, meaning that motion inflated trait-FC effects in some networks but suppressed them in other networks (Supplementary Data 2, 3).

BMI was selected as an exemplar trait to illustrate overestimation of the trait-FC effect due to motion. The trait-FC effects of BMI (FD as a covariate, without motion censoring) were compared to its motion impact score at each edge in Fig. 3 and projected onto the cortical surface using root mean square connectivity[55] in Fig. 4. Many of the network pairs (e.g. on the diagonal: default mode, visual, dorsal attention, and cingulo-opercular) had a negative trait-FC effect and a negative motion impact. Since these trait-FC effects and corresponding motion impacts were both in the same direction (negative in this case), the impact of motion was to inflate the size of the trait-FC effects. An overall motion overestimation score (Stouffer's Z[43]) for BMI of 92 was calculated using omnibus combining across the entire connectivity matrix to control for multiple comparisons. An omnibus significance of p < 0.001 was obtained using permutation.

Note that regressing against the negation of BMI, a trait for which high values were associated with low in-scanner head motion, caused the trait-FC effects of inverse-BMI and the motion impact to both be positive instead of negative. Since the trait-FC effects and corresponding motion impact were still both in the same direction (positive in this counterexample), the overall motion impact score was still a motion overestimation score.

## Residual Motion Can Inflate and Suppress Associations with Demographic and Behavioral Trait

WISC-V matrix reasoning score was selected as an exemplar trait to illustrate a motion underestimation score (Figs. 3 and 4). The overall motion underestimation score for WISC-V of 44 was significant at (omnibus) p < 0.001. The motion overestimation score for WISC-V was not significant. The trait-FC effects for WISC-V (FD as a covariate, without motion censoring) and its respective motion impact scores are shown in Fig. 3 and projected onto the cortical surface in Fig. 4. In networks where the motion impact was significant (e.g. sensorimotor

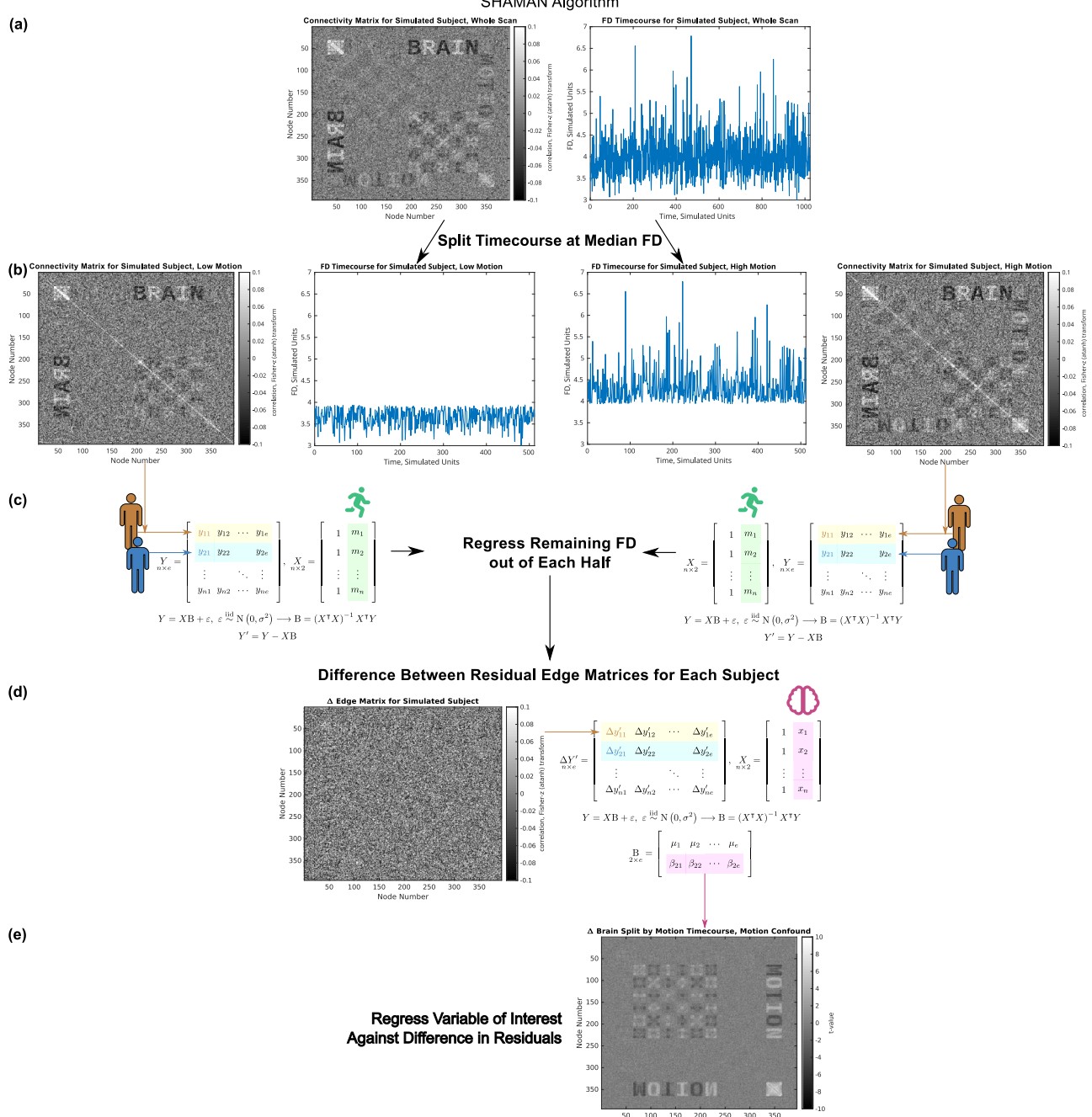

**Fig. 2 | Diagram of the SHAMAN (Split-Half Analysis of Motion-Associated Networks) algorithm using simulated data.** The simulated timeseries of brain signal generates the "BRAIN" edge matrix, and the simulated timeseries of motion generates the "MOTION" edge matrix. **a** A single participant's simulated functional connectivity (FC) matrix and motion (framewise displacement, FD) timeseries. **b** Each participant's fMRI timeseries data is split into high- and low-motion halves using the FD timeseries. High- and low-motion FC matrices and corresponding motion timeseries are shown for the simulated participant. **c** Between-participant variation in motion is regressed out of each half edge matrix to generate residual FC matrices. The difference between the high- and low-motion residual FC matrices is shown for the simulated participant. **d** The trait of interest is regressed against the difference FC matrices. **e** Motion impact score specific to the trait in (**d**).

hand), it trended in the opposite direction (positive vs negative) of the corresponding trait-FC effect, thus suppressing the trait-FC effect.

In total, 34% (13/38) of demographic and behavioral traits had significant motion overestimation scores, and 37% had significant motion underestimation scores, prior to frame censoring (Supplementary Data 2, 3). Broken down further by category, there was a significant motion overestimation score for 20% (1/5) of demographic traits, 50% (5/10) of cognitive traits, and 32% (7/22) of personality traits. There was a significant motion underestimation score for 67% (4/6) of demographic traits, 70% (7/10) of cognitive traits, and 14% (3/22) of

personality traits. Thus, demographic and behavioral traits were less impacted by residual motion than biophysical traits.

## Frame Censoring after Denoising Reduces False Positives due to Motion

Frame censoring is a simple, post-hoc approach to address residual motion artifact after motion processing; however, censoring also has the potential to bias sample proportions through exclusion of high-motion participants[14,20,27,29,31,41]. The SHAMAN algorithm was used to quantify the tradeoff between reduction in motion impact score and

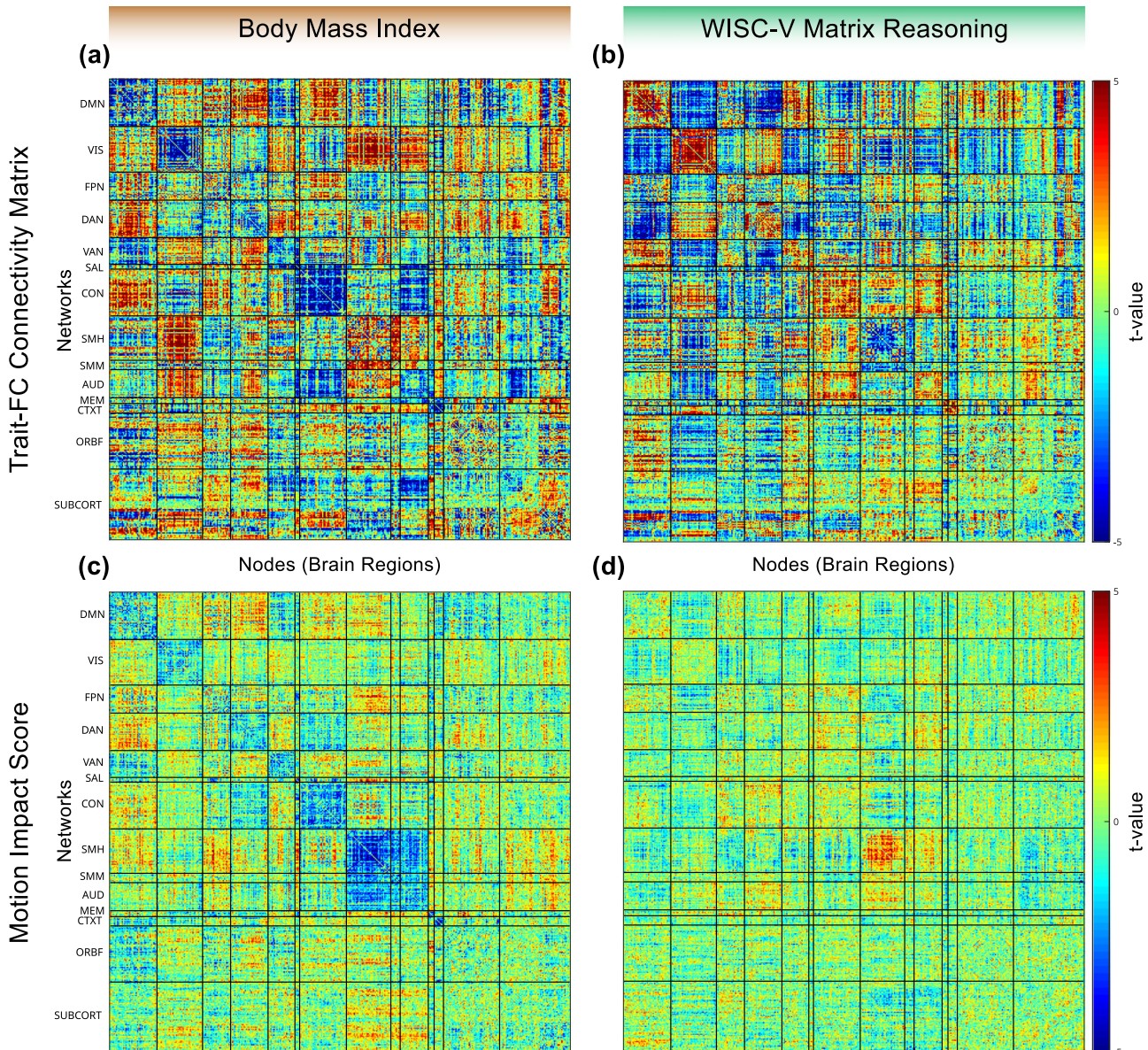

**Fig. 3 | Trait-specific impact of motion on functional connectivity matrices.**
Data are shown for two exemplar traits, body mass index (BMI, left) and WISC-V matrix reasoning score (right). **a** Resting state functional connectivity (RSFC) effect matrix for BMI. **b** Trait-FC effect matrix for matrix reasoning. **c** Motion impact score for BMI. The overall (over the entire matrix, omnibus Stouffer's Z) motion overestimation score for BMI was 92, one-sided $p < 0.001$. **d** Motion impact score for WISC-V. The omnibus motion underestimation score for WISC-V was 44, one-sided $p < 0.001$. All measures were computed after motion processing with ABCD-BIDS and without frame censoring ($n = 7270$). Connectivity matrices were computed with framewise displacement (FD) as a covariate. Networks: DMN Default Mode Network, VIS Visual, FPN Frontoparietal Network, DAN Dorsal Attention Network, VAN Ventral Attention Network, SAL Salience, CON Cingulo-Opercular Network, SMH Somatomotor Hand, SMM Somatomotor Mouth, AUD Auditory, MEM Parietal Memory Network, CTXT Context, ORBF Orbitofrontal, SUBCORT subcortical regions.

sampling bias due to the exclusion of high motion participants (Supplementary Data 2-4). See the work of Pham et al.[41] for additional approaches to optimal censoring threshold selection.

Prior to frame censoring, 7270 participants had at least 8 minutes of rs-fMRI data and were not missing data on any of the 45 trait traits studied. Frame censoring at FD < 0.3 mm (filtered for respiratory motion)[25,33,34] and excluding participants with < 8 minutes of data remaining excluded 5% (384/7270) participants, and censoring at FD < 0.2 excluded 11% (818/7270) of participants. Exclusion of these participants at FD < 0.2 mm caused the average values of only two traits, gender and number of MRI runs completed, to shift by more than 1% (Supplementary Fig. 4, 5) compared to the uncensored sample of participants. On the other hand, censoring at FD < 0.3 reduced the number of traits with significant motion ovestimation scores from 42% (19/45) to 11% (5/45), and censoring at FD < 0.2 further reduced the number to just one physical trait, height, see Fig. 5 (Supplementary Data 2, 3). Similar results using the ABCD data were obtained in our supplementary analyses using DVARS, an alternative to FD for quantifying in-scanner head motion calculated from the root mean square variance of the difference between successive fMRI timepoints. Similar results were obtained in Human Connectome Project (HCP) data where the overall trend of motion impact scores that improved with motion censoring (especially for biophysical traits) was also observed. However, the fraction of traits with significant motion overestimation scores was lower in HCP (9/76, 12%) compared to ABCD (19/45, 42%).

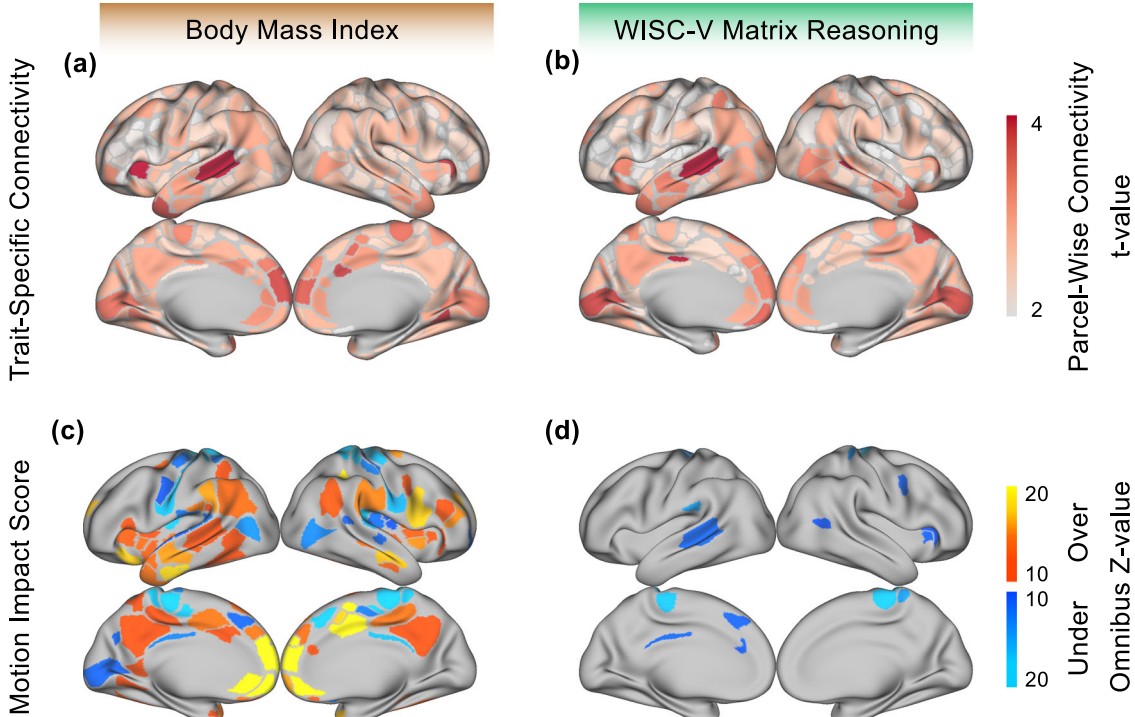

**Fig. 4 | Trait-specific impact of motion on functional connectivity (FC). Top:** Parcel-wise FC, computed as the root mean square (RMS) of connectivity values for each parcel/node in the trait-FC effect matrix[55] for **a** body mass index (BMI) and **b** WISC-V matrix reasoning score. **Bottom:** Motion impact score (omnibus Stouffer's Z[43,71,72], higher = more motion) for **c** BMI and **d** WISC-V. Motion over-estimation scores are labeled "Over" in orange and motion underestimation scores are labeled "Under" in blue. All measures were computed after motion processing with ABCD-BIDS and without frame censoring ($n = 7270$).

More stringent censoring at FD < 0.1 mm (filtered) did not reduce the number of traits in ABCD with significant motion overestimation scores any further, but exclusion of 36% (2712/7270) participants at FD < 0.1 shifted the average values of 24% (11/45) of traits by > 1%, compared to no censoring (Supplementary Data 4, Supplementary Figs. 4 and 5). For example, boys moved their heads 0.07 mm more, on average (mean FD), than girls ($r = 0.10$, $p < 0.001$). At a very stringent motion censoring threshold of FD < 0.1, the study population shifted from majority boys (52.6%) to majority girls (51.6%; Supplementary Fig. 4). Boxplots showing trait distributions at different motion censoring thresholds for all 45 traits can be found in Supplementary Fig. 5. Motion censoring at FD < 0.1 did reduce significant motion underestimation scores from 38% (17/45) with no censoring and at FD < 0.2 to 20% (9/45) at FD < 0.1.

**Motion Impact Was Distributed Across Brain Regions**
Motion impact scores were computed at multiple levels of granularity: for individual pairwise connections (edgewise), combined (using omnibus Stouffer's Z) across edges in a region or node (parcelwise), or combined across the whole brain. Significance at the whole-brain level could theoretically be driven by extremely high motion impact scores concentrated in a few regions or by high motion impact scores distributed across multiple brain regions. To quantify the regional distribution of motion impact score, the whole-brain motion impact score was recalculated after iteratively excluding the brain region with the highest motion impact score until the whole-brain motion impact score was no longer significant. The number of regions excluded was the number of regions significantly contributing to the impact of motion across the brain. We found that, especially for motion over-estimation scores, significant impacts of motion were distributed across many brain regions. BMI and matrix reasoning were selected as exemplar traits to visualize the number of motion-impacted regions needed to detect a significant impact of motion across the whole brain

(Supplementary Fig. 6). For BMI, 39% (155/394) of regions contributed to motion overestimation score whereas <1% of regions contributed to motion underestimation score. For matrix reasoning, 19% (76/394) of regions contributed to motion underestimation score; the whole-brain motion overestimation score for matrix reasoning was not significant.

**Trait-FD Correlation Did Not Predict Motion Overestimation Score**
Calculating motion impact score is computationally intensive and might be unnecessary if overestimation of spurious trait-FC effects could be detected using the trait-motion (trait-FD) correlation alone. A post-hoc analysis was performed to test whether trait-FD correlation predicted motion impact score. Large trait-FD correlation failed to predict motion overestimation score ($r = -0.05$, $p = 0.74$) and therefore did not detect spurious trait-FC correlations without the aid of SHAMAN (Supplementary Fig. 7). This was true even when limiting analysis to behavioral traits, which tend to be of greatest interest to researchers (33 traits, $r = 0.05$, $p = 0.74$). Large trait-FD correlation was correlated with motion underestimation score ($r = 0.36$, $p = 0.015$).

**Data Quantity Affected Motion Impact Score**
Motion impact scores without motion censoring were computed for 7270 participants, of whom 1,291 did not complete all four 5-minute-long resting-state fMRI scans. The number of fMRI frames before censoring was not related to FD overall (Spearman $\rho = 0.155$, $p = 0.20$), but there was a significant negative association when restricting analysis to participants with mean FD > 1.0 mm (Spearman $\rho = -0.188$, $p < 0.001$). Thus, participants with very high amounts of motion had less fMRI data even before motion censoring. A supplementary analysis clamping the number of frames/timepoints to 600 (8 minutes of data) per participant was performed to test whether the ability to detect significant motion impact scores was dependent on participants having unequal amounts of rs-fMRI data. The results are shown

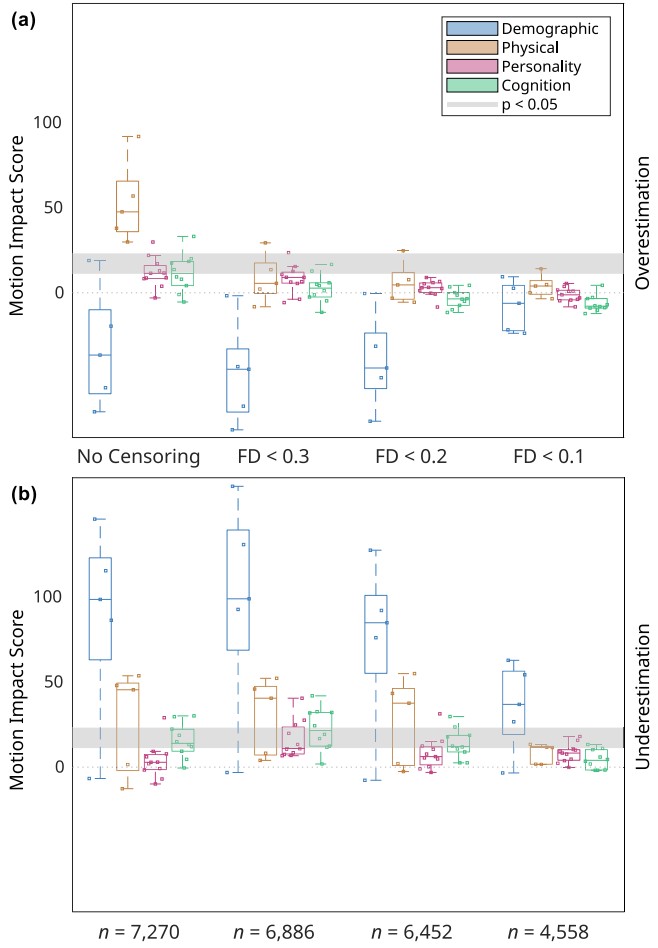

**Fig. 5 | Effects of frame censoring on motion impact score.** Motion impact score (omnibus Stouffer's Z, higher = more motion) for traits of a given category at different levels of motion censoring with $n = 7270$ (no censoring) to $n = 4558$ (strictest censoring) participants. The cutoff score for significance at $p < 0.05$ is different for each trait. The range of significance thresholds is indicated by a gray band. **a** Motion overestimation score. **b** Motion underestimation score. Categories are are demographic (number of people living in household, combined parental income bracket, parents' marital status, highest parental education level, and residential deprivation index), physical (sex assigned at birth, age in months, height, weight, and body mass index), measures from the Child Behavior Check List (CBCL), and measures from the NIH cognitive toolbox. Boxplots indicate the minimum, 25th percentile, median, 75th percentile, and maximum motion impact score.

in Supplementary Data 5. Clamping the number of timepoints to 600, without motion censoring or exclusion of participants, reduced the number of traits with significant motion overestimation scores from 42% (19/45) to 20% (9/45). All 5 of the physical traits still had significant motion overestimation scores after clamping. Clamping the number of frames decreased the number of traits with significant motion underestimation scores from 38% (17/45) to 20% (9/45). Clamping the number of frames was not as effective as motion censoring at FD < 0.2 mm, which reduced the number of traits with significant motion overestimation scores to 2% (1/45).

#### Patterns of Motion Impact Replicate Across Data Sets and Motion Measures

The finding of widespread, trait specific motion impact preferentially affecting primarily biophysical variables and improved by motion censoring is not unique to the ABCD data set. We observed similar patterns of motion impact using an alternative data set, the Human

Connectome Project (HCP); see the Supplementary Material. We also obtained similar results and reduction in motion impact with motion censoring using DVARS[56,57], an alternative to FD for quantifying head motion. The greatest difference between ABCD and HCP was a lower proportion of traits affected by motion overestimation in HCP. Superior robustness to motion impact in the HCP data could be due to using an alternative motion mitigation strategy (ICA-FIX[7,16] for HCP vs ABCD-BIDS for ABCD), differences in participant motion characteristics (FD $0.16 \pm 0.06$ mm for HCP vs $0.29 \pm 0.35$ mm, for ABCD), or differences in available trait measures for each study.

#### Testing for Trait-specific Motion Effects Requires Thousands of Participants

Our analyses were conducted on a large sample ($n = 7270$). We performed bootstrapping (sampling with replacement) at different sample sizes to assess the sensitivity, or statistical power, of SHAMAN to detect significant motion impact scores in our exemplar traits, BMI and matrix reasoning, as a function of sample size. As expected, power increased as the sample size increased (Supplementary Fig. 8). A sample size of about $n = 5000$ participants was powered to detect a motion overestimation score for BMI in 88% of simulations at a significant threshold of ($\alpha = 0.05$). The motion underestimation score for matrix reasoning was more difficult to detect, with successful detection in only 55% of simulations at $n = 5000$. The need for thousands of participants to avoid false positive inference is consistent with recent fMRI studies[49].

#### Motion Impact Score is Specific to Motion-Associated Traits

To be useful, SHAMAN must also be specific, producing not-significant motion impact scores when trait-FC effects are not associated with motion. The randomly-assigned participant ID was used as a trait to simulate a trait-FC effect independent from motion. Participant ID was not significantly correlated with FD ($r = 0.0006$). We performed bootstrapping with participant ID at different sample sizes to assess SHAMAN's specificity. SHAMAN did not falsely detect a significant motion impact score at any sample size from 100 to 6,000 participants (Supplementary Fig. 8).

### Discussion

#### The Impact of Motion on Brain-Wide Association Studies is Widespread and Trait Dependent

It has been proposed that head motion likely affects inference about some trait-FC effects more than others[6,19,27-30], but only a few studies[12,18] have explored the differential impact of motion on specific trait-FC effects within a statistical framework. We build upon this prior work[18] by employing a novel metric, motion impact score, to quantify the impact of residual in-scanner head motion on trait-FC effects for 45 demographic, biophysical, intelligence, and personality traits within the ABCD study. Similar to prior findings in the Human Connectome Project (HCP) data[18], we find that, even after denoising with ABCD-BIDS, a large proportion (42%) of trait-FC effects in the ABCD data ($n = 7270$) are significantly inflated by residual head motion when motion censoring is not used, and that post-hoc motion censoring is an effective strategy for mitigating residual motion.

#### Motion Impact May Arise from Heterogeneous Sources

The SHAMAN method distinguishes between transient (state) and persistent (trait) changes in fMRI signal related to a nuisance time-series (head motion), but it does not distinguish between different biophysical sources of transient fMRI signal. Such sources could include not only artifactual signal from proton spin effects, but also fMRI activation in motor cortex, and even the transition between different FC states[58]. Researchers specifically studying the relationship between small head movements and transient changes in neuronal activity might employ our SHAMAN method to discover interesting FC

changes related to transient head motion. For more conventional BWAS, any fMRI signal change time-locked with head motion, regardless of source, has the potential to affect reproducibility if not removed or controlled for.

## Motion May Negatively Affect the Signal-to-Noise-Ratio (SNR)

We find that, after denoising, the motion-FC effect matrix has a large effect size (max $|r| = 0.10$) and is strongly anti-correlated with the between-participant average FC matrix ($r = -0.63$), see Fig. 1 and Supplementary Fig. 10, thus potentially impacting the interpretation of many traits whose FC effects are shaped by the average FC matrix. One parsimonious explanation for this motion-connectivity anti-correlation is that data from participants with higher motion have a lower signal to noise ratio (SNR). Decreased SNR could occur due to the biophysical characteristics of head motion increasing the proportion of noise relative to brain signal captured by the MRI scanner. We also observe that participants with very high motion (mean FD > 1.0 mm) completed fewer fMRI scans leading to a shorter duration of fMRI data, higher sampling variability, and thus lower SNR. It is also possible that denoising algorithms inadvertently remove more brain signal from high-motion data than they do from low-motion data. Designers of denoising algorithms may wish to evaluate not only how much motion is removed from the data, but also how the SNR of the data are affected by processing.

While we hypothesize that motion-related signal loss plays a large role in explaining the similarity between the motion-FC effect matrix and average FC matrix, we cannot definitely prove the association from these analyses. Other factors contributing to the observed motion-FC effect matrix may include distance-dependent attenuation of connectivity by motion artifact[40]. Propensity of an individual to move is itself a stable, heritable trait[59–61], and some[62] have even proposed that intrinsic differences in FC related to movement are encoded in the FC matrix. While our findings do not exclude the possibility that motion has an intrinsic neurobiological basis, we find it surprising that motion's trait-FC effect would have a larger effect size than any other trait-FC effect. It is more plausible that the motion-FC effect arises from a combination of artifact, signal loss, and biologically-meaningful motion-related traits and FC states.

## Biophysical Traits Were Most Likely to be Confounded by Motion Distortions

While we find that many trait-FC relationships (42%) are spuriously inflated by motion, it is remarkable that all five biophysical trait-FC relationships (BMI, age, sex, weight, and height) are significantly inflated by motion. Biophysical traits are also the only class for which we detect significant motion underestimation scores after very stringent censoring at FD < 0.1. As seen in Fig. 3 illustrating BMI, the largest deviations in the trait-FC connectivity matrices for biophysical traits cluster along the diagonal and have a large degree of similarity to both the average (across participants) FC matrix and the motion-FC effecty matrix. It is possible that the biophysical traits are directly related to motion (e.g. larger head sizes are capable of greater displacement), affect signal to noise ratio in a similar way to motion (e.g. excess adipose tissue attenuates radiofrequency energy, decreasing SNR), or that biophysical traits serve as a proxy for other behavioral traits related to movement. If the relationship between biophysical traits and SNR or behavioral covariates could be modeled then their true trait-FC relationships could be identified.

## Motion Censoring Reduces the Risk of False Positive Inference

Motion censoring at FD < 0.2 mm (filtered for respiratory artifact) is very effective at mitigating motion overestimation score (Fig. 5, Supplementary Data 2, 3) and results in relatively little selection bias due to participant exclusion (Supplementary Fig. 4, 5, Supplementary Data 4) in this neurotypical cohort. Siegel et al.[18] also observed decreased residual impact of motion after framewise censoring in the HCP data. Less stringent censoring at 0.3 retained more participants but also more traits with motion overestimation; even more stringent censoring at FD < 0.1 (filtered) does not further reduce motion overestimation score, has little effect on motion underestimation score, and introduces greater selection bias, for example, reversing the ratio of boys to girls in the censored data compared to the uncensored data. The degree of selection bias would likely be even greater for children with developmental disorders such as autism[31]. Therefore, motion censoring at FD < 0.2 mm (filtered) is a reasonable strategy to avoid making false positive inferences about most trait-FC effects.

The potential for selection bias[31,40] and loss of power from motion censoring might be further addressed through a combination of approaches including: recruiting more participants from high-motion groups (e.g. children with autism), scanning them for longer, adopting behavioral strategies to reduce in-scanner head motion[22], employing real-time adaptive quality assurance strategies such as FIRMM[5] to ensure collection of sufficient data to maintain the desired power after frame censoring, adaptively selecting an optimal motion censoring threshold[41], and employing statistical methods to account for selection bias[31].

## Motion Reduces Statistical Power

Statistical power is the ability to reject the null hypothesis when it is actually false, or the ability to avoid false negative inference. We show that motion predisposes to false negative inference about trait-FC effects (motion underestimation score) for many traits, even after motion censoring. Therefore, motion reduces statistical power for specific traits in a way that is not accounted for by conventional power analysis methods. While researchers may be more attentive to false positive results, false negative results and statistical power are also important. For example, failure to detect an association between lead exposure and neurotoxicity would contribute to incorrect policy decisions about lead mitigation, increased lead exposure, and adverse public health outcomes. False negative results are especially damaging when they arise in large brain wide associations studies (BWAS, e.g., ABCD), which are assumed to be adequately powered.

## SHAMAN Facilitates Region of Interest (ROI) Analyses

It is common to select a priori regions or nodes of interest on which to perform FC analyses. ROI analyses may also be beneficial for excluding spatially-structured motion artifacts. Using non-parametric combining, SHAMAN can assign a motion impact score to a single node, a subset of nodes, or the whole brain, reflecting a continuum[44] from high specificity (at the level of the ROI) and low power to low specificity and high power (across the whole brain) to detect motion impact. ROI analysis of motion impact score can provide insight into whether a priori regions of interest are impacted by residual motion, and facilitate analysis of a few nodes, selected a priori, which do not have a significant motion impact score even when the whole brain does have a significant motion impact score. While ROI analysis is typically used to limit the number of statistical tests or comparisons, as the brain can be subdivided into thousands of combinations of ROIs, the number of possible statistical tests/comparisons is vast. Therefore, caution is advised when interpreting "cherry picked" regions of interest *a posteriori* with a low motion impact score, especially for traits where motion impact score is widely distributed over a large proportion of nodes (e.g., BMI: 39% of nodes).

## SHAMAN Can Quantify Statistical Significance of Motion Distortion

The problem of estimating motion artifact is intractable because the precise relationship between head motion and fMRI signal is not known[26]. Denoising algorithms must therefore make assumptions about the motion-fMRI relationship. For example, regressing the

motion parameters out of the fMRI signal[29] makes a strong, linear assumption about the motion-fMRI relationship. Alternative machine learning approaches such as ICA-FIX[7,16] may achieve greater robustness to motion artifact by relaxing these linear assumptions, but no method is assumption-free. Detecting residual motion after denoising is especially challenging because the same assumptions used by the denoising algorithm cannot be reused. For example, regressing the motion parameters out of the fMRI signal a second time will not remove additional motion artifact. SHAMAN approaches this problem by assuming that the high-motion half of the data will have the same correlation structure as the low-motion half except for artifact from motion. Therefore, while quantifying residual motion precisely is impossible, SHAMAN detects when the impact of motion on a trait-FC relationship is significant. In theory the residual impact of motion might be significantly non-zero but small enough that it would not cause false inference. However, consistent with prior work[49], we show that the motion-FC effect is similar in magnitude to the effect size of most trait-FC effects. Therefore, a significant motion overestimation score is likely to cause false inference.

### SHAMAN Predicts Residual Motion Impact Better Than Trait-motion Correlation Alone

This study was motivated by the hypothesis that trait-motion (FD) correlations for traits (e.g. BMI) correlated with head motion would be more likely to be spuriously related to motion. Are strong trait-FD correlations sufficient to predict spurious trait-FC effects? Traits highly correlated with motion were more likely to have significant motion underestimation scores. However, strong trait-FD correlations did not predict which trait-FC relationships would be spuriously inflated by motion, biasing toward false positive results. Therefore, SHAMAN provides vital information about the risk of false-positive inference due to residual trait-specific motion impact beyond the trait-motion correlation alone.

### Detecting the Impact of Residual Trait-Specific Motion Requires Thousands of Participants

Marek et al.[49] showed that thousands of participants are needed to measure trait-FC effects reproducibly. Similarly, we find that 2000–3000 participants are needed to detect the motion over-estimation score on the BMI-FC effect, and 6000 participants are needed to detect the motion underestimation score on the matrix reasoning-FC effect (Supplementary Fig. 8). We postulate that motion underestimation scores require larger sample sizes for detection because affected trait-FC effects would tend to have smaller effect sizes. Our findings therefore reinforce the benefit of large sample sizes for BWAS. When large samples are not feasible (e.g. in studies of rare diseases), extrapolation of our findings suggests framewise motion censoring is a prudent strategy to avoid false positive inferences. Researchers should also exercise caution when interpreting trait-FC effects for biophysical traits (e.g. BMI) because they are particularly susceptible to spurious inflation by motion artifact.

### Quantifying Motion Impact Helps To Avoid False Inference

Many traits of interest to human population neuroscience are significantly correlated with head motion (87%). Motion denoising algorithms successfully remove much of the effect of motion, but trait-specific motion effects remain likely to distort inferences drawn from functional connectivity. The risk of false positives is highest for biophysical traits (e.g., BMI) and false negatives are most likely for cognitive/behavioral traits (e.g. matrix reasoning). Frame censoring is an effective strategy to reduce trait-specific motion distortions and can increase the likelihood of finding a significant, reproducible effect. Methods for finding an optimal frame censoring threshold[41] can be combined with SHAMAN to minimize data loss. Spatial masks generated by our novel SHAMAN method can be used to avoid the most motion-impacted regions during ROI selection. Given that residual head motion is most likely to falsely suppress true associations between FC and cognitive traits, more aggressive motion suppression and denoising techniques should lead to the discovery of new cognition-FC relationships.

## Methods

### Standard Methods

**Ethics.** This project used resting-state functional MRI, demographic, biophysical, and behavioral data from 11,572 9–10 year old participants from the ABCD 2.0 release[63]. The ABCD Study obtained centralized institutional review board (IRB) approval from the University of California, San Diego. Each of the 21 sites also obtained local IRB approval. Ethical regulations were followed during data collection and analysis. Parents or caregivers provided written informed consent, and children gave written assent.

**Behavioral.** The Adolescent Brain Cognitive Development (ABCD) study participants are well-phenotyped with demographic, physical, mental health[47], and cognitive[48] batteries. For the purpose of this report we selected TR 45 broadly-interesting traits for which complete data was available on a majority of participants: NIH Toolbox, Wechsler Intelligence Scale for Children (WISC-V)[53], Cognitive Behavioral Checklist (CBCL), Prodromal Psychosis Scale (PPS), Behavioral Inhibition/Avoidance Scales (BIS/BAS), and Urgency Premeditation Perseverance Sensation Impulsive Scale (UPPS-P), see Supplementary Data 1. We also selected 5 physical (BMI, age, sex, weight, and height) and 5 demographic (number of people living in household, combined parental income bracket, parents' marital status, highest parental education level, and residential deprivation index) traits commonly used as covariates, as well as 2 study-related variables (number of MRI scans completed and drowsiness), to see if any of these would be highly correlated with motion. Data were downloaded from the NIMH Data Archive (ABCD Release 2.0), and the traits of interest were extracted using the ABCDE software we have developed and which we have made available here: https://gitlab.com/DosenbachGreene/abcde.

**Exemplar Traits.** Two exemplar traits were selected from the ABCD study *a posteriori* to illustrate the SHAMAN method. Body mass index (BMI) is measured in units of kg/m² with a normative range of about 14–22 for children ages 9–10 years[54]. The Wechsler Intelligence Scale for Children, 5th edition (WISC-V) matrix reasoning subscore has a mean of 10, standard deviation of 3, and maximum value of 19[53], with higher scores indicating superior performance on the test.

**MR Imaging.** Functional magnetic resonance imaging (fMRI) was acquired at 21 sites using a protocol harmonized for 3 Tesla GE, Philips, and Siemens scanners with multi-channel receive coils[21]. In addition to anatomical and task-fMRI, each participant had up to four 5-minute-long resting-state scans (TR = 800 ms, 20 min total). Participants with less than 8 min of resting-state data, the minimum duration needed for high-quality estimation of connectivity[64], were excluded from analysis. A subset of sites using Siemens scanners used FIRMM motion tracking software[5] that allows extending the scan on the basis of on-line measurement of motion.

Following acquisition, fMRI data were processed using standardized methods including correction for field distortion, frame-by-frame motion co-registration, alignment to standard stereo-tactic space, and extraction of the cortical ribbon[39]. Resting-state data were further processed to remove respiratory and motion artifact by temporal bandpass filtering, global signal regression, and regression against the rigid-body motion parameters using the ABCD-BIDS motion processing pipeline, a derivative of the Human Connectome Project (HCP) processing pipeline[38] described by Fair et al.[25,37]. Processing dependencies include FSL[65] and FreeSurfer[66].

**Parcellation.** It is possible to compute functional connectivity between each voxel or vertex. However, this approach is burdened by a high proportion of unstructured noise and large computer memory requirements. We therefore adopted a ROI-based approach based on the 333 cortical parcels described by Gordon et al.[67] augmented by the 61 subcortical parcels described by Seitzman et al.[68] for a total of 394 parcels, or nodes.

**Quantifying Motion.** Motion in fMRI studies is typically estimated using spatial co-registration of each fMRI volume (or frame) to a reference frame, or temporal analysis of variance[14]. In this study we quantify motion using framewise displacement, FD (L1-norm), in mm after filtering for respiratory artifact[33,34] because it is the default metric used in the ABCD-BIDS processing pipeline[25,37].

**Assessing the Performance of ABCD-BIDS.** The primary aim of this study was to quantify *trait-specific* residual motion, but we also quantified trait-agnostic residual motion for comparison to our trait-specific findings and to assess the overall performance of the ABCD-BIDS motion processing tool (Supplementary Fig. 1). Multiple methods have been developed to quantify trait-agnostic residual motion after the fMRI data have been denoised or "processsed."[40] We employed a straightforward approach of measuring the proportion of between-participant variation in fMRI signal variance explained by motion. FD was averaged over each participant's resting-state fMRI scans. The fMRI signal variance within each parcel was computed for each participant before and after ABCD-BIDS (without frame censoring). The fMRI signal variance was averaged across parcels to generate a single measure of fMRI signal variance for each participant. The relationship between fMRI signal variance and framewise displacement was fit with a log-log model. Assumptions of linearity were further relaxed by using the square of Spearman's $\rho$ instead of the coefficient of determination $R^2$. The relative reduction in the proportion of signal variance related to motion achieved by ABCD-BIDS was computed as 1 - $(\rho_{after}/\rho_{before})^2$.

**Correlation of Demographic & Behavioral Traits with Motion.** We compare continuous traits of interest with motion (FD, in mm) using the linear product-moment correlation coefficient, r. This quantity reflects the degree to which in-scanner head motion is related to a trait of interest and is reported in Supplementary Fig. 2.

**Frame Censoring.** Frame censoring excludes spurious variance at the cost of data loss[69]. Therefore, determining the optimal frame censoring threshold is an empirical question. We considered the effect of no motion censoring (i.e. all frames included in analysis), censoring at an FD cutoff of 0.3 mm, censoring at a stringent FD cutoff of 0.2 mm, and censoring at a very stringent FD cutoff of 0.1 mm. Participants with less than 8 min of resting-state data were excluded from further analysis at each censoring threshold, including the no motion censoring threshold. This resulted in a different number of participants at each censoring threshold (Supplementary Fig. 4). The 8 min cutoff was selected due the difficulty of estimating stable values for functional connectivity with shorter scan duration[64].

**Functional Connectivity.** We begin with the standard approaches for computing and making inferences about functional connectivity, which are familiar to all fMRI researchers. To avoid ambiguity, the methods are briefly summarized here, and diagrammed in Supplementary Fig. 9 for reference. By convention, each brain region or parcel is referred to as a "node" and the functional connections between nodes, which are referred to as "edges," are computed as the pairwise linear correlation coefficients between nodes. A vector of edges is typically visualized in an edge, correlation, effect, or "connectivity" matrix. As correlations are constrained to vary from −1 to 1, the correlation coefficients are atanh (FIsher Z) transformed prior to

regression. Inferences about trait-FC effects are made by modeling the atanh-transformed edges as a linear combination of some participant-wise trait of interest (e.g. matrix reasoning score) and some covariates (e.g. FD) using least-squares regression. The estimate of the trait-FC effect at each edge is divided by the standard error of the estimate to obtain a t-value at each edge.

## Methods for SHAMAN

**Rationale.** We begin by assuming *traits* and their effect on resting-state functional connectivity are stable over time whereas *states* vary with time[70]. Here a trait is a variable such as height, intelligence score, or favorite color, which is stable over the timescale of an fMRI scan. Extrinsic variables like parental income are also considered traits for this purpose. State is a variable such as instantaneous respiratory rate or head motion that varies during an fMRI scan. Although it has been observed that the propensity to move is a trait which is stable over months to years[60], here we consider second-to-second variations in the FD timeseries.

It follows from this assumption that if we split an individual participant's fMRI BOLD timeseries in half, we will draw the same individual- and group-level inferences on a trait from each half of the data (to within sampling error). If trait-specific connectivity is not associated with head motion then split-half inference will be invariant to the amount of motion in each half of the data. Thus, we can detect trait-specific motion-associated connectivity by comparing inferences when the data is split according to the FD timeseries (see Supplementary Fig. 13 for distribution of median FD) vs split at random.

**Algorithm Steps.** The SHAMAN (**S**plit-**H**alf **A**nalysis of **M**otion-**A**ssociated **N**etworks) algorithm is diagrammed in Fig. 2. Its steps are enumerated below.

1. An individual participant's resting-state BOLD timeseries from one or more fMRI scans are concatenated (Fig. 2a).
2. The individual participant's fMRI BOLD timeseries is split in half (Fig. 2b). One half (left) contains the half of the timeseries with the lowest FD (motion) values, and the other half (right) contains the half with the highest FD values.
3. A connectivity/edge matrix is computed separately for each half of the individual participant's data (Fig. 2b). See Supplementary Fig. 10 for an illustration using real connectivity data.
4. Average (over time) FD is computed for each half of the data (Fig. 2c).
5. Steps 1-4 are repeated for each participant in the study.
6. The between-participant effect of motion is accounted for by regressing (across participants) mean FD out of each half of the connectivity data (Fig. 2c).
7. The difference in residuals between low- and high-motion halves from Step 6 is computed for each participant (Fig. 2d).
8. The trait/variable of interest is regressed against the difference in residuals from Step 7 (Fig. 2d), optionally in the presence of covariates.
9. The resultant connectivity matrix is the motion impact for that trait (Fig. 2e).

If motion processing works perfectly, or if participants do not move at all, then each half of the data will yield identical connectivity matrices, and the difference in residuals in Step 7 above (Fig. 2d) will be zero to within sampling error. Subsequently, the result of regression in Steps 8 and 9 will also be zero (Fig. 2e).

If spurious effects of motion remain in the data then the difference in residuals in Step 7 (Fig. 2d) will not be all zeros. If the trait/variable of interest is associated with head motion then the regression in Steps 8 and 9 will reveal the trait-specific motion impact (Fig. 2e).

**Statistical Inference.** The SHAMAN algorithm reveals the trait-specific impact of motion. This connectivity matrix will be zero under the null

hypothesis of zero motion impact. To detect significant motion impact, we test the omnibus alternative hypothesis that one or more edges in this matrix is non-zero. To avoid making strong parametric assumptions about the data, we use a permutation scheme followed by the non-parametric combining method described by Winkler et al.[43].

The permutation scheme is diagrammed in Supplementary Figs. 11 and 12 and described in greater detail in the following section. In Step 2 of the algorithm (Supplementary Fig. 11b) the data are split in half at random, without respect to the FD timeseries. The rest of the algorithm proceeds as before. The algorithm is repeated many times with different random splits (permutations) of the data to obtain a null distribution for the motion-associated connectivity matrix (Supplementary Fig. 11e).

Once a null distribution is generated, we use non-parametric combining[43] to test the null hypothesis that all edges in the motion-associated connectivity matrix are zero. Each edge is treated as a separate "modality" to avoid assumptions about exchangeability or clustering of edges. We selected Stouffer's combining function[71,72] because Stouffer's Z-score is well-known and has an intuitive interpretation. The non-parametric combining process yields a single, omnibus Stouffer's Z-score for the entire connectivity matrix, which we call the motion impact score. Comparing the motion impact score to its null distribution yields an omnibus p-value that controls for family-wise error rate across the entire motion-associated connectivity matrix. When the p-value is small a significant amount of motion-associated connectivity is present. When the motion impact score (Stouffer's Z-score) is large, a large amount of motion associated connectivity is present.

**Motion Blocks Permutation.** Functional MRI timeseries are known to be weakly non-stationary and autocorrelated[73], therefore the assumption of exchangeability and unconstrained permutation of timepoints may not be appropriate. Specifically, consecutive timepoints could be randomly assigned to opposite halves of the data, making it more likely that the randomly (during permutation) split halves of the timeseries data would be similar. This would have the effect of making the difference between the randomly split halves smaller, thereby shrinking the null distribution toward zero and overestimating the significance of the motion impact score.

The problem of inference on weakly-dependent data is well-known in the statistical literature. Politis & Romano[74] describe the block and stationary bootstrap procedures for use on weakly-dependent data. Essentially, rather than resample individual timepoints with replacement, the Politis & Romano procedure resamples blocks of consecutive timepoints with replacement, where the block lengths are drawn from a geometric distribution. However, resampling the data into randomly-sized blocks makes it difficult to optimally separate the high- and low-motion halves of the data, since the randomly-sized blocks might not conform to the natural divisions of the motion timeseries.

Therefore, we devised a scheme we call "motion blocks" permutation that allows for optimal separation of high- and low-motion data while accounting for weak temporal dependency within the timeseries. The motion blocks permutation scheme is depicted in Supplementary Fig. 12. For the un-permuted split, the data are divided into high- and low-motion halves as usual, resulting in "blocks" of consecutive high- and low-motion data points. For the permuted split, the blocks, rather than individual data points, are randomly assigned to each half of the data. This strategy preserves the same amount of weak temporal dependence in the permuted data as in the un-permuted data.

**The SHAMAN p-value Relates to Motion, Not Broader Significance.** The p-value generated by the SHAMAN algorithm tests the omnibus null hypothesis that no edge (pairwise connection) in a variable/trait's FC effect matrix is significantly obscured by or inflated by head motion. It does *not* test the hypothesis that the variable itself has a significant relationship with connectivity. On the contrary, a small p-value from SHAMAN (or a correspondingly large motion score) indicates that variable's relationship with FC may be spuriously associated with head motion.

**Motion Overestimation vs Underestimation Score.** The main objective of SHAMAN is to detect how motion impacts trait-FC effect for specific traits. When motion impact has the same sign as the trait-FC effect (both positive or both negative) then motion causes overestimation of the size of the trait-FC effect and would tend to cause false positive inference (concluding the trait-FC effect is significant when it is not). When motion impact has the opposite sign of the trait-FC effect then motion causes underestimation of the size of the trait-FC effect and would tend to cause false negative inference (failing to detect a significant trait-FC effect when it really is significant). We can then compute separate motion impact scores, a motion overestimation score and a motion underestimation score, by counting only those edges whose motion impact is the same (overestimation) or opposite (underestimation) sign as the trait-FC effect. Distinguishing between over- and under-estimation requires us to know with some degree of certainty whether the trait-FC effect is positive or negative. We use a heuristic approach in which the trait-FC effect at a given edge is positive if its t-value is > 2 and negative if its t-value is < −2. If the t-value is near zero then we ignore the motion impact at that edge.

**Timeseries Simulation.** In order to test the validity of the SHAMAN algorithm we devised a generative model for fMRI BOLD timeseries data for which the parameters of brain signal and motion can be experimentally controlled. Such a model necessarily requires simplifying assumptions since the true biophysical properties of the brain are not known. Several fMRI data simulators exist[75], but none allows for realistic simulation of head motion artifact. Therefore, for our simulator, we worked backwards from the assumptions of the conventional, massively univariate, resting-state functional connectivity regression model (Supplementary Fig. 9) in which between-participant variables of interest are predictors (the design matrix) and atanh (Fisher Z) transformed linear correlation coefficients between brain regions are the observed data.

We generated two basis correlation matrices with the same dimensions (394 × 394 nodes) as the Gordon-Laumann-Seitzman[67,68] brain parcellation containing bitmap representations of the text "brain" and "motion," see Fig. 2 and Supplementary Fig. 14. The Cholesky decomposition of each basis correlation matrix was used to simulate the timeseries of brain and motion signals. The process for generating these simulated timeseries is diagrammed in Supplementary Fig. 15. Normally distributed random data were multiplied by the Cholesky decomposition of the respective basis correlation matrix to generate simulated "BRAIN" and "MOTION" timeseries for each simulated participant. We simulated 394 nodes x 1024 fMRI frames (time points) for 1024 simulated participants. The variance, across the 394 nodes, at each of 1024 time points in the motion timeseries was used as the FD timeseries for each participant.

The brain and motion timeseries were mixed together in controlled proportions. Working backwards, the 1024 participant x 1 predictor "brain" column of the across-participants design matrix (i.e. the trait variable) was drawn at random from the standard normal distribution and then shifted to obtain all positive numbers. The 1024 × 1 "motion" column of the design matrix was taken as the mean FD timeseries of each participant. The brain and motion timeseries for each participant were then mixed by addition in the time domain according to their proportions (brain, motion) of variance in the design matrix. For example, if a participant's row in the design matrix were (1.6329, 0.2216) then the mixed timeseries was generated by mixed = sqrt(1.6329) * brain + sqrt(0.2216) * motion. An additional step detailed in the Supplementary

Material was used to correct the variance of the mixed timeseries to satisfy the assumptions of linear regression on correlation matrices.

**Node Analysis.** We can quantify motion impact within individual nodes/parcels of the brain network. This may be advantageous when planning a region of interest analysis or to help visualize motion-associated connectivity in anatomical space. Instead of performing omnibus non-parametric combining over the entire connectivity matrix, we perform non-parametric combining to obtain a Stouffer's Z-score and *p*-value over the edges connected to a single node or subset of nodes.

The process for node analysis is best illustrated In Supplementary Fig. 6. First, an omnibus *p*-value is computed across all edges. Then a Stouffer's Z-score (motion impact score) is computed for each individual node using the edges connected to that node. The node with the largest motion score is excluded by deleting its connecting edges from the connectivity matrix. The whole process is then repeated until all nodes have been excluded. The thick red/black line in Supplementary Fig. 6 shows the omnibus *p*-value increasing (becoming less significant) from left to right as each successive node is excluded. Portions of the line below $p = 0.05$ are highlighted in red. The point at which the line intersects with $p = 0.05$ corresponds to the number of nodes affected by motion-associated connectivity.

Node analysis is also used to visualize motion-associated connectivity anatomically in Fig. 4c,d. The Stouffer's Z-score, or "motion Z-score," is computed for each individual node as above. The resultant motion scores are visualized on the cortical surface.

### Reporting summary
Further information on research design is available in the Nature Portfolio Reporting Summary linked to this article.

## Data availability
Participant level data from ABCD are openly available pursuant to consortium-level data access rules. The ABCD data repository grows and changes over time (https://nda.nih.gov/abcd). The ABCD data used in this study came from ABCD Annual Release 2.0 (https://doi.org/10.15154/1503209).

## Code availability
Analysis code specific to this study can be found at https://github.com/DosenbachGreene/shaman. Code for processing ABCD and UKB data can be found at https://github.com/DCAN-Labs/abcd-hcp-pipeline.

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

## Acknowledgements

ABCD Acknowledgement Data used in the preparation of this article were obtained from the Adolescent Brain Cognitive Development (ABCD) Study (https://abcdstudy.org), held in the NIMH Data Archive (NDA). This is a multisite, longitudinal study designed to recruit more than 10,000 children age 9-10 and follow them over 10 years into early adulthood. The ABCD Study is supported by the National Institutes of Health and additional federal partners under award numbers U01DA041022, U01DA 041028, U01DA041048, U01DA041089, U01DA041106, U01DA041117, U01DA041120, U01DA041134, U01DA041148, U01DA041156, U01DA 041174, U24DA041123, U24DA041147, U01DA041093, and U01DA041025. A full list of supporters is available at https://abcdstudy.org/federal-partners.html. A listing of participating sites and a complete listing of the study investigators can be found at https://abcdstudy.org/scientists/workgroups/. ABCD consortium investigators designed and implemented the study and/or provided data but did not necessarily participate in analysis or writing of this report. This manuscript reflects the views of the authors and may not reflect the opinions or views of the NIH or ABCD consortium investigators. The ABCD data repository grows and changes over time. The ABCD data used in this report came from Anual Release 2.0, https://doi.org/10.15154/1503209. This work was supported by NIH grants MH100019 (SM), MH121518 (SM), MH123091 (AZ), NS115672 (AZ), NS133486 (JLR), DA057486 (BT-C), MH129616 (TOL), NS110332 (DJN), MH100019 (NAS), MH129493 (DMB), NS123345 (BPK), NS098482 (BPK), NS098577 (AZS), DA041148 (DAF), DA04112 (DAF), MH115357 (DAF), MH096773 (DAF and NUFD), MH122066 (EMG, DAF, and NUFD), MH121276 (EMG, DAF, and NUFD), MH124567 (EMG, DAF, and NUFD), NS129521 (EMG, DAF, and NUFD), NS140256 (EMG and NUFD), and NS088590 (NUFD); by the National Spasmodic Dysphonia Association (EMG); by Mallinckrodt Institute of Radiology pilot funding (EMG); by the McDonnell Center for Systems Neuroscience (AZ); by the Taylor Family Institute Fund for Innovative Psychiatric Research (TOL); by the Andrew Mellon Predoctoral Fellowship from the Dietrich School of Arts & Sciences, University of Pittsburgh (BTC); and by the Extreme Science and Engineering Discovery Environment (XSEDE) Bridges at the Pittsburgh Supercomputing Center through allocation TG-IBN200009 (BTC).

## Author contributions

B.P.K. is the corresponding author. B.P.K., D.F.M., S.F.M., B.T.C., J.S.S., B.A., T.O.M., A.M., R.J.C., A.N.V., S.R.K., D.J.N., A.Z., N.A.S., E.F., A.R., O.M.D., J.L.R., E.M.G., A.Z.S., D.M.B., D.A.F., and NUFD made substantial contributions to the conception and design of the analysis. R.L.M., K.M.S., J.M., L.A.M., A.J.P., G.M.C., E.A.E., S.M.M., M.C., O.D., B.J.L., J.C.W., T.P., and A.M.G. made substantial contributions to data acquisition and analysis. VS made substantial contributions to the SHAMAN software.

## Competing interests

D.A.F. and N.U.F.D. have a financial interest in Turing Medical Inc. and may financially benefit if the company is successful in marketing FIRMM motion-monitoring software products. D.A.F. and N.U.F.D. may receive royalty income based on FIRMM technology developed at Washington University School of Medicine and Oregon Health and Sciences University and licensed to Turing Medical Inc. D.A.F. and N.U.F.D. are co-founders of Turing Medical Inc. These potential conflicts of interest have been reviewed and are managed by Washington University School of Medicine, Oregon Health and Sciences University and the University of Minnesota. The other authors declare no competing interests.

## Additional information

[1]Department of Neurology, Washington University School of Medicine, St. Louis, MO, USA. [2]Department of Psychiatry, Washington University School of Medicine, St Louis, MO, USA. [3]Mallinckrodt Institute of Radiology, Washington University School of Medicine, St. Louis, MO, USA. [4]Masonic Institute for the Developing Brain, University of Minnesota, Minneapolis, MN, US. [5]Department of Psychiatry and Behavioral Sciences, University of Minnesota, Minneapolis, MN, USA. [6]Institute for Translational Neuroscience, University of Minnesota, Minneapolis, MN, USA. [7]Department of Psychiatry, NYU Langone Medical Center, New York, NY, USA. [8]Department of Biomedical Engineering, Washington University in St Louis, St Louis, MO, USA. [9]Department of Neurology, NYU Langone Medical Center, New York, NY, USA. [10]Department of Pediatrics, University of Minnesota, Minneapolis, MN, USA. [11]Data Science and Sharing Team, National Institute of Mental Health, NIH, DHHS, Bethesda, MD, USA. [12]Department of Psychology, University of Minnesota, Minneapolis, MN, USA. [13]Joint Doctoral Program in Clinical Psychology, University of California San Diego, San Diego, CA, USA. [14]Department of Psychiatry, Oregon Health & Science University, Portland, OR, USA. [15]Minnesota Supercomputing Institute, University of Minnesota, Minneapolis, MN, USA. [16]University of Minnesota Informatics Institute, University of Minnesota, Minneapolis, MN, USA. [17]Taylor Family Department of Neurosurgery, Washington University School of Medicine, St. Louis, MO, USA. [18]Department of Psychological and Brain Sciences, Washington University in Saint Louis, St Louis, MO, USA. [19]Institute of Child Development, University of Minnesota, Minneapolis, MN, USA. [20]Department of Pediatrics, Washington University School of Medicine, St Louis, MO, USA. ✉e-mail: benjamin.kay@wustl.edu

