## [Transparent Peer Review file · Nature Communications]

Motion Impact Score for Detecting Spurious Brain-Behavior Associations

Corresponding Author: Dr Benjamin Kay

This manuscript has been previously reviewed at another journal. This document only contains information relating to versions considered at Nature Communications. Mentions of the other journal have been redacted.

Version 0:

Reviewer comments:

Reviewer #1

(Remarks to the Author)

I previously reviewed this paper when it was submitted to [REDACTED]. Therefore, this review will focus on my previous concerns and my evaluation of the paper today in light of those concerns. In my earlier review, I outlined 5 major concerns. While the authors have made changes to address 3 of those concerns, there are 2 major concerns from my earlier review that remain substantial weaknesses of the paper.

Briefly, there are two things that this paper does not currently do, which I believe if done successfully could make the paper a “home run”. First, it would clearly establish that all differences between lower-motion (FD below the median) and higher-motion (FD above the median) periods in an fMRI session are entirely undesirable and should be eliminated. That is, there is no true signal that may actually be biologically relevant and desirable to retain. Second, once the desired goal is established and defended (remove all differences between lower/higher motion periods? or just those differences related to artifacts and motor activation?), a comprehensive comparison of motion mitigation strategies would determine the most effective method at achieving that goal. Those two things correspond to the following two concerns from my previous review.

1. In my earlier review, one of my major concerns was “Not checking the major assumption underlying the proposed SHAMAN method”, where I stated the following:

In addition, a major assumption underlying the proposed SHAMAN method is left unverified: from the discussion, the “the weak but reasonable assumption that the high-motion half of the data will have the same correlation structure as the low-motion half except for artifact from motion.” Is this assumption really reasonable? It seems to me completely plausible that periods during which a subject is moving represent a different FC state than periods during which a subject is holding still. For example, periods of high and low motion could represent periods during which the DMN is more or less active. This assumption should be thoroughly examined, and the limitations should be acknowledged, rather than ignoring the possibility that the assumption might be too strong.

In response to that concern, the authors made two changes: (a) adding a supplementary figure (Fig. S10), and (b) adding a paragraph to the discussion. I will now comment on both of these changes:

a) The new supplementary figure (Fig. S10) shows the mean FC matrix during lower-motion and higher-motion halves, along with the difference between them. As far as I can tell, this new figure is not mentioned or interpreted in the paper (there is language in the paper referring to Fig. S10 but it appears to be in reference to a different figure), and the authors do not offer any interpretation in their response to reviewers. Visually inspecting the figure, one notices substantial differences in connectivity between the DMN and three other networks: FPN, DAN, and CON. Connectivity or anti-connectivity between the DMN and attention/control networks is related to focus and attention. Therefore, this figure suggests biologically plausible differences in FC between lower-motion halves and higher-motion halves of the data. There are many other differences to be seen in the figure, and it's not clear which of these are due to (1) actual artifact or motion per-se (e.g. motor activation) vs. (2) other FC features that may represent the propensity to spend time in different cognitive states. While removal of the first source of differences is clearly desirable, removal of the second source of differences may be

undesirable. That brings us to the second change made by the authors.

b) To motivate the new discussion paragraph, the authors in their response to reviewers state the following:

“One axiomatic position proffered by the reviewer holds that artifact is spurious signal induced by the MRI scanner. Our axiomatic assumption is that any signal that would affect inference is an “artifact” in the statistical sense, even if that signal arises from a real neurobiological process such as motor cortex activation, FC state, etc.”

Likewise, in the new discussion paragraph, they state that

“SHAMAN [...] does not distinguish between different biophysical sources of transient fMRI signal. Such sources could include not only artifactual signal from proton spin effects, but also fMRI activation in motor cortex, and even the transition between different FC states.”

This is quite a strong statement. The authors are not claiming that there are no real differences in FC between higher and lower motion halves of the data. Based on Fig. S10, indeed there likely are real FC state differences between higher and lower motion halves. What is remarkable is that SHAMAN sees such FC differences as undesirable, indeed a form of “artifact”. If my understanding is correct, the authors acknowledge that motion scrubbing may alter the subject-specific distribution of FC states. Rather than view this as a limitation and a reason to proceed with caution, this is seen as desirable. In this reviewer’s opinion, it is not desirable to remove or down-weight real FC states from the data. To allay this potential concern, it would be necessary to back up this position with evidence. For instance – one can imagine many associations between behavior and the propensity to spend time in certain FC states associated with higher motion. The authors could try to show that conversely, standardizing the distribution of FC states across subjects actually somehow improves FC-behavior prediction. However, currently this strong assertion is not defended with any analyses.

One last thought on this point – I am guessing the proposed motion impact score metrics would continue to decrease to some minimal level as the FD threshold approaches the median value in the scan. That is, the higher-motion half of the data follows some distribution, ranging from $\text{med}(\text{FD})$ to $\text{max}(\text{FD})$. As the FD threshold for motion scrubbing approaches $\text{med}(\text{FD})$, the differences between the higher/lower halves of the data should be minimized, as long as sufficient data is retained in the higher half to have sufficient data to compute the metric. Therefore, minimizing this metric seems to be perhaps not the right way to use it. Is there perhaps a more nuanced way to use the motion impact score metric to assess the impact of scrubbing? Perhaps focusing on particular types of connections known to be most impacted by scrubbing, and looking for where the score plateaus?

2. Another concern from my earlier review was “Inadequate comparison of methods to censor motion artifact”, where I stated:

To provide a full picture of the pros and cons of different levels and approaches to censoring, the authors should consider at least two alternative methods: ICA-FIX, which has been shown to have a similar effect as motion scrubbing due to its ability to remove spikes in the time series; and data-driven scrubbing, such as the most recent version of DVARS (Afyouni and Nichols 2018, Neuroimage).

In response to that concern, the authors added supplementary figures SB.2 (ICA-FIX) and SB.3. (DVARS). However, these other methods really do not feature in the main analyses of the paper, which is primarily focused on motion scrubbing using FD. These supplementary analyses are described as follows in the paper:

“Similar results using the ABCD data were obtained in our supplementary analyses using DVARS, an alternative to FD for quantifying in-scanner head motion calculated from the root mean square variance of the difference between successive fMRI timepoints. The overall trend of significant motion impact scores without motion censoring that improved with motion censoring at $\text{FD} < 0.2$ mm (especially for biophysical traits) was also observed in data from the Human Connectome Project (HCP).”

Regarding ICA-FIX, censoring with $\text{FD} < 0.2$ shows a slight reduction for some trait categories, but it’s not clear if those differences are significant. By contrast, without ICA-FIX the results of motion scrubbing are dramatic (Fig. 5). ICA-FIX is never compared side-by-side or in the text with motion scrubbing, and the separation of the figures corresponding to each makes such comparison difficult for readers. Regarding DVARS, it appears from Fig. SB.3 to be effective at reducing motion impact scores, but again it is not directly compared with FD, and the separation between the figures and differences in y-axis scale make comparison difficult for the reader. In short, these other methods are now included in the supplement, but the paper does not seek to compare different methods. Likewise, more lenient FD thresholds often used in practice (e.g. $0.2\text{mm} < \text{FD} < 0.5\text{mm}$) are not considered.

Two brief additional comments regarding the DVARS analysis: First, I couldn’t find many details about the DVARS implementation, but based on Fig. SB.3 it appears that a cutoff of 200, rather than the statistically principled dual cutoff Afyouni and Nichols (2018), was adopted. Since that principled cutoff technique was found to be superior, it would be preferable to adopt it. Second, the amount of volumes removed via DVARS vs. FD is not reported, it seems, which is relevant information when comparing scrubbing methods.

Finally, the authors partially addressed another major concern from my earlier review:

3. “The use of statistical methods that assume linearity without verification”: The authors now report Spearman rank

correlations, which do not depend on an assumption of linearity. However, I find it problematic that Pearson correlations are still reported, which is not appropriate when the data clearly exhibit non-linear associations. On a related point, it is clear from some of the plots that some measures exhibit high degrees of right skew and should probably be log-transformed prior to fitting lines through them. For example, both x- and y-variables in Fig. S1a, and the y-variable in Fig. S2.

I wish the authors well in this work, and I hope they find these perspectives helpful.

(Remarks on code availability)

Reviewer #3

(Remarks to the Author)

The resubmission has been substantially responsive to prior reviews and is substantively improved. However, there are two areas which were identified by Reviewer #1 in the first reviews which still require addressing:

1. "Our axiomatic assumption is that any signal that would affect inference is an "artifact" in the statistical sense, even if that signal arises from a real neurobiological process such as motor cortex activation, FC state, etc." I disagree that this should be called an "artifact", a term that implies it is not a real biological effect in common usage. The authors need to acknowledge this is a short-coming of the proposed method (that real connectivity differences could be tied to motion increases/decreases) and avoid calling it an artifact.
2. Linearity is still assumed in most models (Pearson R, linear correlation, simulations). There is no need to assume this, as there are plenty of non-linear regression methods that could be employed (e.g., random forests).

(Remarks on code availability)

N/A

Version 1:

Reviewer comments:

Reviewer #1

(Remarks to the Author)

I appreciate the effort the authors have made to respond to my earlier feedback. The Discussion section especially is now more nuanced and precise. However, I find that there remain several matters of clarity and communication, including some issues raised in my previous review. While I still have several concerns, they should be fairly straightforward to address:

1. The question of motion artifact versus real motion state is now addressed in the Discussion, but in many places the word "artifact" is still used to refer to all time-varying changes in FC associated with motion. This is potentially misleading, since as the authors acknowledge, there may be real changes in FC that occur during periods of higher motion. Replacing "motion artifacts" with, say, "motion effects" or "motion impacts" would acknowledge that, as the authors say in the discussion, "motion impact may arise from heterogeneous sources" including artifacts, activation, and FC states.

Similarly, the wording "between-participant differences due to head motion" is too strong, as it implies a purely causal relationship between head motion and FC. A better word choice would be "differences associated with head motion" to acknowledge the possibility that head motion may also be indirectly associated with FC due to the influence of various motion-associated traits.

2. In my last review, I requested that the authors present DVARS alongside FD, but the DVARS results are still only presented in the appendix. This is highly inconvenient for readers who wish to consider DVARS alongside FD. The authors should combine Figures 5 and SB.3 (ABCD), as well as Figures SB.2 and SB.4 (HCP). In addition, comparing these two pairs of figures, the "No Censoring" panel should be the same, I assume, across the corresponding figures, but it appears to differ. For example, the left-most group of boxplots in Figure 5 do not appear to match the left-most group of box plots in Figure SB.3. Is there a mistake? (Minor issue: in Figure SB.4 the older terminology of "False Positive" and "False Negative" are used.) In order to facilitate the consideration of DVARS by readers, it is important to present the DVARS results alongside the FD results in the main figures (Fig 5 for ABCD, Fig SB.4 for HCP).

3. Regarding DVARS, the authors explain that they use the "normalized" version of Afyouni and Nichols (2019), but if I understand correctly, they do not utilize the principled cutoff technique. Afyouni and Nichols proposed two complementary versions of DVARS and a technique for generating the null distribution of each measure, which can then be thresholded at a quantile to identify abnormal volumes with high specificity. One advantage of this approach is that it is invariant to the TR of the data, and hence generalizes across different datasets like the ABCD and HCP. It is important to utilize this improved

version of DVARS with the principled cutoff, rather than applying an ad-hoc cutoff.

4. In response to my previous feedback, the authors now use a permutation scheme that preserves temporal dependencies. This is vital in order to satisfy the assumption of exchangeability in permutation testing. Unfortunately, results based on “unconstrained permutation”, which violates the assumption of exchangeability, are still presented. There is no guarantee of correct false positive control with a permutation procedure that breaks the underlying structure of the data. Unconstrained permutation is not a valid permutation procedure in this setting, and as such it should be fully replaced by block permutation that preserves temporal autocorrelations.

5. The title and text of the final results section “Motion Impact Score is Specific to Motion Artifact”, are potentially misleading. The analysis in this section is based on a “fake” non-trait, namely participant ID, which is not related to motion. This analysis does not establish whether Motion Impact Score is “specific to motion artifact” for realistic traits that are themselves associated with motion. Also, it is unclear why $n=100$ participants was considered sufficient for this analysis, while the previous section established that $n = 5,000$ subjects are needed to successfully detect motion overestimation score (and even that many are not sufficient to detect motion underestimation score). My main concern, however, is that the analysis in this section is designed to determine whether Motion Impact Score produces false positives for a null trait, but it cannot determine whether it produces false positives for a more realistic trait. In addition, the wording “specific to motion artifact” implies that SHAMAN can distinguish between motion artifact and motion-associated signal. It is important to reword both the title and text of this section to accurately describe the conclusions that can be drawn from this particular analysis.

6. The third section of the Discussion, titled “Motion May Negatively Affect the SNR”, includes a claim that is not clearly justified. The authors first observe that motion-FC associations are themselves anti-correlated with FC. Then, they say that “the most parsimonious explanation for this motion-connectivity anti-correlation is that data from participants with higher motion have a lower SNR”. I don’t see how this follows. Another possible interpretation of the data could be that weaker functional connections (low values of FC) are more susceptible to motion (high values of motion-FC association). If the authors are interested in investigating the relationship between motion and SNR, that could be interrogated more directly. Perhaps the authors's current claim about SNR is correct, but if so, it requires a more clear and convincing explanation.

7. The analyses in this manuscript provide many interesting insights, but most of the conclusions are based on the ABCD. The ABCD has obvious advantages (namely its size) but also limitations due to the specific nature of the study population, data acquisition methods, and data processing/denoising methods. It is not clear how well the conclusions made here generalize. Comparison with the HCP only partly alleviates this limitation, because (1) the results differ quite a lot, comparing Figures 5 and SB.2, and (2) as explained by the authors, there are a number of differences between the datasets, making it difficult to pin down the effects of population, processing, and other factors. Comparing Figures 5 and SB.2, it seems clear that the “no censoring” strategy in the HCP is much less problematic than in the ABCD. The most likely explanation for this is that the HCP uses a fundamentally different denoising strategy (ICA-FIX), which could be more effective at mitigating motion artifact. I do not expect the authors to tease out all of the reasons for the differences between the ABCD and HCP results. However, it is important to (1) acknowledge the lower magnitude and frequency of motion impacts in HCP (the Discuss section “Patterns of Motion Impact Replicate Across Data Sets” currently understates the differences) and (2) acknowledge as a limitation that the main analyses in the paper rely on specific denoising strategies, and that motion impacts and the impacts of frame censoring may be lessened when using ICA-FIX (this is supported by the HCP results) or other advanced denoising strategies like multi-echo ICA.

8. The stated assumption that “the high-motion half of the data will have the same correlation structure as the low-motion half except for artifact from motion” is not “weak” in my opinion, and it is in conflict with other parts of the discussion that describe the diverse potential drivers of motion-FC effects (i.e. “artifact, signal loss, and biologically-meaningful motion-related traits and FC states”). Therefore, it seems necessary to revisit the wording of this assumption.

9. In the Frame Censoring section of the Methods, the new FD cutoff of 0.3 is not mentioned, nor is DVARS.

I believe that this feedback should be straightforward to address, and am hopeful that the authors will find it helpful.

(Remarks on code availability)

Response to Reviewers

We the authors appreciate the insightful feedback from Reviewer #1 and the opportunity to revise and further strengthen the manuscript. In response to the reviewer's important comments, we incorporated key supplemental findings into the main message of the manuscript. The revised manuscript includes an improved method for estimating DVARS and additional FD thresholds as requested by the reviewer.

The reviewer's comments are highlighted gray. Our comments are in blue text, and excerpts from the manuscript are indented with **changes highlighted in yellow**.

Reviewer #1

Remarks to the Author

I previously reviewed this paper when it was submitted to [REDACTED]. Therefore, this review will focus on my previous concerns and my evaluation of the paper today in light of those concerns. In my earlier review, I outlined 5 major concerns. While the authors have made changes to address 3 of those concerns, there are 2 major concerns from my earlier review that remain substantial weaknesses of the paper.

Briefly, there are two things that this paper does not currently do, which I believe if done successfully could make the paper a "home run". First, it would clearly establish that all differences between lower-motion (FD below the median) and higher-motion (FD above the median) periods in an fMRI session are entirely undesirable and should be eliminated. That is, there is no true signal that may actually be biologically relevant and desirable to retain. Second, once the desired goal is established and defended (remove all differences between lower/higher motion periods? or just those differences related to artifacts and motor activation?), a comprehensive comparison of motion mitigation strategies would determine the most effective method at achieving that goal. Those two things correspond to the following two concerns from my previous review.

We greatly appreciate the potential "home run" impact of our approach to head motion. The reviewer has identified two remaining major concerns for use to address. First is the question of whether differences in fMRI signal time-locked to head motion are always undesirable, or whether some such signals might have biological significance in the appropriate context. We have expanded the Discussion to address contexts in which biological interpretation of motion-related fMRI signals would be more or less appropriate.

Second, the reviewer requests a "comparison of motion mitigation strategies." The revised manuscript now discusses data how ICA-FIX likely has superior robustness to motion artifact compared to ABCD-BIDS, and we agree that a more comprehensive comparison of motion

mitigation strategies would be most useful to neuroimaging research, but it is unfortunately beyond the scope of this current manuscript.

Patterns of Motion Impact Replicate Across Data Sets

The finding of widespread, trait specific motion impact preferentially affecting primarily biophysical variables and improved by motion censoring is not unique to the ABCD data set. We observed similar patterns of motion impact using an alternative data set, the Human Connectome Project (HCP). The greatest difference between ABCD and HCP was a lower proportion of traits affected by motion overestimation in HCP. Superior robustness to motion artifact in the HCP data could be due to using an alternative motion mitigation strategy (ICA-FIX^{7,16} for HCP vs ABCD-BIDS for ABCD), differences in participant motion characteristics (FD 0.16 ± 0.06 mm for HCP vs 0.29 ± 0.35 mm, for ABCD), or differences in available trait measures for each study.

SHAMAN Can Quantify Statistical Significance of Motion Distortion

The problem of estimating motion artifact is intractable because the precise relationship between head motion and fMRI signal is not known²⁶. Denoising algorithms must therefore make assumptions about the motion-fMRI relationship. For example, regressing the motion parameters out of the fMRI signal²⁹ makes a strong, linear assumption about the motion-fMRI relationship. Alternative machine learning approaches such as ICA-FIX^{7,16} may achieve greater robustness to motion artifact by relaxing these linear assumptions, but no method is assumption-free...

1. In my earlier review, one of my major concerns was “Not checking the major assumption underlying the proposed SHAMAN method”, where I stated the following:

In addition, a major assumption underlying the proposed SHAMAN method is left unverified: from the discussion, the “the weak but reasonable assumption that the high-motion half of the data will have the same correlation structure as the low-motion half except for artifact from motion.” Is this assumption really reasonable? It seems to me completely plausible that periods during which a subject is moving represent a different FC state than periods during which a subject is holding still. For example, periods of high and low motion could represent periods during which the DMN is more or less active. This assumption should be thoroughly examined, and the limitations should be acknowledged, rather than ignoring the possibility that the assumption might be too strong.

SHAMAN does not distinguish between fMRI signal changes arising from motion-induced artifacts in the fMRI acquisition process vs biologically-driven changes in fMRI signal time-locked with the head motion timecourse. We have revised the following paragraph in our discussion to make this point even clearer.

Motion Impact May Arise from Heterogeneous Sources

The SHAMAN method distinguishes between transient (state) and persistent (trait) changes in fMRI signal related to a nuisance timeseries (head motion), but it does not distinguish between different biophysical sources of transient fMRI signal. Such sources could include not only artifactual signal from proton spin effects, but also fMRI activation

in motor cortex, and even the transition between different FC states⁵⁶. Researchers specifically studying the relationship between small head movements and transient changes in neuronal activity might employ our SHAMAN method to discover interesting FC changes related to transient head motion. For more conventional BWAS, any fMRI signal change time-locked with head motion, regardless of source, has the potential to affect reproducibility if not removed or controlled for.

In response to that concern, the authors made two changes: (a) adding a supplementary figure (Fig. S10), and (b) adding a paragraph to the discussion. I will now comment on both of these changes:

a) The new supplementary figure (Fig. S10) shows the mean FC matrix during lower-motion and higher-motion halves, along with the difference between them. As far as I can tell, this new figure is not mentioned or interpreted in the paper (there is language in the paper referring to Fig. S10 but it appears to be in reference to a different figure), and the authors do not offer any interpretation in their response to reviewers. Visually inspecting the figure, one notices substantial differences in connectivity between the DMN and three other networks: FPN, DAN, and CON. Connectivity or anti-connectivity between the DMN and attention/control networks is related to focus and attention. Therefore, this figure suggests biologically plausible differences in FC between lower-motion halves and higher-motion halves of the data. There are many other differences to be seen in the figure, and it's not clear which of these are due to (1) actual artifact or motion per-se (e.g. motor activation) vs. (2) other FC features that may represent the propensity to spend time in different cognitive states. While removal of the first source of differences is clearly desirable, removal of the second source of differences may be undesirable.

We greatly appreciate the reviewer's in-depth analysis of our revised figures. Figure S10 was added as an explication of Figure 1, but we provided insufficient context for interpretation of this new figure. We now reference Figure S10 in the caption of Figure 1 and in the corresponding paragraph in the Discussion to provide greater context.

Motion May Negatively Affect the Signal-to-Noise-Ratio (SNR)

We find that, after denoising, the motion-FC effect matrix has a large effect size (max $|r| = 0.10$) and is strongly anti-correlated with the between-participant average FC matrix ($r = -0.63$), see Figures 1 and S10, thus potentially impacting the interpretation of many traits whose FC effects are shaped by the average FC matrix. The most parsimonious explanation for this motion-connectivity anti-correlation is that data from participants with higher motion have a lower signal to noise ratio (SNR)...

As previously shown, other factors contributing to the observed motion-FC effect matrix may include... Propensity of an individual to move.... While our findings do not exclude the possibility that motion has an intrinsic neurobiological basis, it seems surprising that motion's trait-FC effect would have a larger effect size than any other trait-FC effect. It seems more plausible that the motion-FC effect arises from a combination of artifact, signal loss, and biologically-meaningful motion-related traits and FC states.

Upon visual inspection, we agree with the reviewer that Figures 1(c)/S10(c) include large off-diagonal changes in connectivity of the DMN with FPN, DAN, and CON. However, these off-diagonal changes are also present in the average connectivity matrix, see Figure 1(a)/S10(a). Quantitative analysis shows anti-correlation (Spearman $\rho = -0.5822$) between Figures 1(c)/S10(c) and the average connectivity matrix involving all of the canonical networks. Thus, while these off-diagonal differences involving attention/control networks are intriguing, we do not feel comfortable interpreting them in our data. A follow-up study with a behavioral/task intervention, such as asking participants to hold more or less still, might provide greater clarity about the role of these executive networks as they relate to head motion.

Figure 1: Average connectivity matrix, motion-FC effect matrix, and effect sizes. (a) The average (across participants) correlation matrix for functional connectivity (FC), after denoising with ABCD-BIDS without frame censoring, see Figure S10 to visualize average FC matrices from high- and low-motion halves of the data separately; (b) and with frame censoring at framewise displacement (FD) < 0.2 mm. (c) The motion-FC effect matrix obtained by regressing average FD against FC for each participant, $FC \sim 1 + FD$. The motion-FC effect matrix has units of change in FC per mm FD. (d) Motion-FC effect matrix with censoring at FD < 0.2. (a,c) The upper triangular parts of the FC connectivity matrices were edge-for-edge correlated at Pearson $r = -0.6256$ (Spearman $\rho = -0.5822$). (b,d) After motion censoring at FD < 0.2 mm, the average FC and motion-FC effect matrices were edge-for-edge correlated at Pearson $r = -0.5517$ (Spearman $\rho = -0.5059$). (e) The effect size of the motion-FC effect (arrow) is plotted relative to the empirical distribution of effect sizes for the 45 traits in this study without motion censoring and (f) with motion censoring at framewise displacement (FD) < 0.2 mm. Effect sizes were computed as the largest (98th percentile) normalized difference in functional connectivity (correlation, $|r|$) across edges for each trait. Networks: DMN = Default Mode Network, VIS = Visual, FPN = Frontoparietal Network, DAN = Dorsal Attention Network, VAN = Ventral Attention Network, SAL = Salience, CON = Cingulo-Opercular Network, SMH = Somatomotor Hand, SMM = Somatomotor Mouth, AUD = Auditory, MEM = Parietal Memory Network, CTXT = Context, ORBF = Orbitofrontal, SUBCORT = subcortical regions.

That brings us to the second change made by the authors.

b) To motivate the new discussion paragraph, the authors in their response to reviewers state the following:

“One axiomatic position proffered by the reviewer holds that artifact is spurious signal induced by the MRI scanner. Our axiomatic assumption is that any signal that would affect inference is an “artifact” in the statistical sense, even if that signal arises from a real neurobiological process such as motor cortex activation, FC state, etc.”

Likewise, in the new discussion paragraph, they state that

“SHAMAN [...] does not distinguish between different biophysical sources of transient fMRI signal. Such sources could include not only artifactual signal from proton spin effects, but also fMRI activation in motor cortex, and even the transition between different FC states.”

This is quite a strong statement. The authors are not claiming that there are no real differences in FC between higher and lower motion halves of the data. Based on Fig. S10, indeed there likely are real FC state differences between higher and lower motion halves. What is remarkable is that SHAMAN sees such FC differences as undesirable, indeed a form of “artifact”. If my understanding is correct, the authors acknowledge that motion scrubbing may alter the subject-specific distribution of FC states. Rather than view this as a limitation and a reason to proceed with caution, this is seen as desirable. In this reviewer’s opinion, it is not desirable to remove or down-weight real FC states from the data. To allay this potential concern, it would be necessary to back up this position with evidence. For instance – one can imagine many

associations between behavior and the propensity to spend time in certain FC states associated with higher motion. The authors could try to show that conversely, standardizing the distribution of FC states across subjects actually somehow improves FC-behavior prediction. However, currently this strong assertion is not defended with any analyses.

Motion scrubbing and antecedent motion processing/mitigation could indeed alter the subject-specific distribution of FC states. The necessity of such imperfect methods for fMRI analysis is not a novel finding in our manuscript, but is a well-known finding established in numerous prior studies (Power 2013, Siegel 2017, Power 2018, etc). In contrast to prior work, our SHAMAN method allows for quantification of residual motion impact specific to a phenotypic trait or hypothesis. This specificity could allow for a more liberal scrubbing threshold than would nominally be permitted – over half of the traits in ABCD were usable without any scrubbing – and thus inclusion of more motion-specific FC states than would have previously been possible.

The reviewer makes a thoughtful suggestion to show that standardizing the distribution of FC states across participants improves FC-behavior prediction. This topic is addressed by contemporary work in one of the reviewer's suggested references, Pham et al 2023 (<https://doi.org/10.1016/j.neuroimage.2023.119972>), who showed that there is an optimal scrubbing threshold that balances data quality and loss of predictive power. In our revised discussion, we make multiple suggestions to combine Pham's method for finding an optimal scrubbing threshold with our method for quantifying the residual impact of motion on FC-behavior relationships after scrubbing.

Frame Censoring after Denoising Reduces False Positives due to Motion Artifact

Frame censoring is a simple, post-hoc approach to address residual motion artifact after motion processing; however, censoring also has the potential to bias sample proportions through exclusion of high-motion participants^{14,20,27,29,31,41}. The SHAMAN algorithm was used to quantify the tradeoff between reduction in motion impact score and sampling bias due to the exclusion of high motion participants (Tables S2-S4). See the work of Pham et al⁴¹ for additional approaches to optimal censoring threshold selection.

The potential for selection bias^{31,40} and loss of power from motion censoring might be further addressed through a combination of approaches including: recruiting more participants from high-motion groups (e.g. children with autism), scanning them for longer, adopting behavioral strategies to reduce in-scanner head motion,²² employing real-time adaptive quality assurance strategies such as FIRMM⁵ to ensure collection of sufficient data to maintain the desired power after frame censoring, adaptively selecting an optimal motion censoring threshold⁴¹, and employing statistical methods to account for selection bias³¹.

Quantifying Motion Impact Helps To Avoid False Inference

Many Traits of interest to human population neuroscience are significantly correlated with head motion (87%). Motion denoising algorithms successfully remove much of the effect of motion, but trait-specific motion effects remain likely to distort inferences drawn from functional connectivity. The risk of false positives is highest for biophysical traits (e.g., BMI) and false negatives are most likely for cognitive/behavioral traits (e.g. matrix

reasoning). Frame censoring is an effective strategy to reduce trait-specific motion distortions and can increase the likelihood of finding a significant, reproducible effect. **Methods for finding an optimal frame censoring threshold⁴¹ can be combined with SHAMAN to minimize data loss.** Spatial masks generated by our novel SHAMAN method can be used to avoid the most motion-impacted regions during ROI selection. Given that residual head motion is most likely to falsely suppress true associations between FC and cognitive traits, more aggressive motion suppression and denoising techniques should lead to the discovery of new cognition-FC relationships.

One last thought on this point – I am guessing the proposed motion impact score metrics would continue to decrease to some minimal level as the FD threshold approaches the median value in the scan. That is, the higher-motion half of the data follows some distribution, ranging from $med(FD)$ to $max(FD)$. As the FD threshold for motion scrubbing approaches $med(FD)$, the differences between the higher/lower halves of the data should be minimized, as long as sufficient data is retained in the higher half to have sufficient data to compute the metric. Therefore, minimizing this metric seems to be perhaps not the right way to use it. Is there perhaps a more nuanced way to use the motion impact score metric to assess the impact of scrubbing? Perhaps focusing on particular types of connections known to be most impacted by scrubbing, and looking for where the score plateaus?

We believe the reviewer is correct that, with extremely stringent motion censoring, the motion impact score would asymptotically approach zero. One way the reviewer suggests to use the motion impact score, would be to select a scrubbing threshold that minimizes motion impact score within an *a priori* neuroanatomically-defined mask or region of interest. This is now mentioned in the Discussion (see above). We also provide a permutation-based statistical inference framework to assess the significance of motion impact score over the whole connectome or within a set of *a priori* edges of interest. We suggest targeting a not-significant motion overestimation score, which mitigates the risk of false-positive inference, rather than trying to minimize the motion impact score itself.

2. Another concern from my earlier review was “Inadequate comparison of methods to censor motion artifact”, where I stated:

To provide a full picture of the pros and cons of different levels and approaches to censoring, the authors should consider at least two alternative methods: ICA-FIX, which has been shown to have a similar effect as motion scrubbing due to its ability to remove spikes in the time series; and data-driven scrubbing, such as the most recent version of DVARS (Afyouni and Nichols 2018, Neuroimage).

In response to that concern, the authors added supplementary figures SB.2 (ICA-FIX) and SB.3. (DVARS). However, these other methods really do not feature in the main analyses of the paper, which is primarily focused on motion scrubbing using FD. These supplementary analyses are described as follows in the paper:

“Similar results using the ABCD data were obtained in our supplementary analyses using DVARS, an alternative to FD for quantifying in-scanner head motion calculated from the root mean square variance of the difference between successive fMRI timepoints. The overall trend of significant motion impact scores without motion censoring that improved with motion censoring at $FD < 0.2$ mm (especially for biophysical traits) was also observed in data from the Human Connectome Project (HCP).”

Regarding ICA-FIX, censoring with $FD < 0.2$ shows a slight reduction for some trait categories, but it's not clear if those differences are significant. By contrast, without ICA-FIX the results of motion scrubbing are dramatic (Fig. 5). ICA-FIX is never compared side-by-side or in the text with motion scrubbing, and the separation of the figures corresponding to each makes such comparison difficult for readers. Regarding DVARS, it appears from Fig. SB.3 to be effective at reducing motion impact scores, but again it is not directly compared with FD, and the separation between the figures and differences in y-axis scale make comparison difficult for the reader. In short, these other methods are now included in the supplement, but the paper does not seek to compare different methods.

Reviewer #1 suggested a comparison of methods to censor motion artifact, specifically the inclusion of DVARS (as a motion measure) and ICA-FIX (as a preprocessing method). As suggested, we added these additional analyses to the revised manuscript. It appears that in HCP data processed with ICA-FIX there is greater robustness to motion artifact compared to ABCD data processed with the ABCD-BIDS pipeline. We have added this observation to our Results:

The fraction of traits with significant motion overestimation scores was lower in HCP (9/76, 12%) compared to ABCD (19/45, 42%).

And in our discussion:

Patterns of Motion Impact Replicate Across Data Sets

The finding of widespread, trait specific motion impact preferentially affecting primarily biophysical variables and improved by motion censoring is not unique to the ABCD data set. We observed similar patterns of motion impact using an alternative data set, the Human Connectome Project (HCP). The greatest difference between ABCD and HCP was a lower proportion of traits affected by motion overestimation in HCP. Superior robustness to motion artifact in the HCP data could be due to using an alternative motion mitigation strategy (ICA-FIX^{7,16} for HCP vs ABCD-BIDS for ABCD), differences in participant motion characteristics ($FD 0.16 \pm 0.06$ mm for HCP vs 0.29 ± 0.35 mm, for ABCD), or differences in available trait measures for each study.

Reviewer #1 also observed that data for HCP/ICA-FIX and DVARS were presented in the supplemental materials. We have made additional references to the supplemental material, including a new subheading without our Discussion (see above). Demonstrating that SHAMAN replicates in data outside ABCD and with measures other than FD makes our work more

impactful. However, the manuscript's primary focus is to introduce a novel method for quantifying trait-specific residual motion impact, therefore including too many additional analyses outside of this primary focus could reduce the clarity of the manuscript. We agree with the reviewer that further, comprehensive comparisons of motion mitigation measures and scrubbing strategies will be an exciting direction for future work.

Likewise, more lenient FD thresholds often used in practice (e.g. $0.2\text{mm} < \text{FD} < 0.5\text{mm}$) are not considered.

The revised manuscript now includes an FD threshold of 0.3 mm.

Prior to frame censoring, 7,270 participants had at least 8 minutes of rs-fMRI data and were not missing data on any of the 45 trait traits studied. Frame censoring at $\text{FD} < 0.3$ mm (filtered for respiratory motion)^{25,33,34} and excluding participants with < 8 minutes of data remaining excluded 5% (384/7270) participants, and censoring at $\text{FD} < 0.2$ excluded 11% (818/7270) of participants. Exclusion of these participants at $\text{FD} < 0.2$ mm caused the average values of only two traits, gender and number of MRI runs completed, to shift by more than 1% (Figures S4, S5) compared to the uncensored sample of participants. On the other hand, censoring at $\text{FD} < 0.3$ reduced the number of traits with significant motion overestimation scores from 42% (19/45) to 11% (5/45), and censoring at $\text{FD} < 0.2$ further reduced the number to just one physical trait, height, see Figure 5 (Tables S2, S3).

Figure 5: Effects of frame censoring on motion impact score. Motion impact score (omnibus Stouffer's Z, higher = more motion) for traits of a given category at different levels of motion censoring. The cutoff score for significance at $p < 0.05$ is different for each trait. The range of significance thresholds is indicated by a gray band. **(a)** Motion overestimation score. **(b)** Motion underestimation score. Categories are demographic (number of people living in household, combined parental income bracket, parents' marital status, highest parental education level, and residential deprivation index), physical (sex assigned at birth, age in months, height, weight, and body mass index), measures from the Child Behavior Check List (CBCL), and measures from the NIH cognitive toolbox. Boxplots indicate the minimum, 25th percentile, median, 75th percentile, and maximum motion impact score.

Variable	SHAMAN p-Value											
	No Censoring			FD < 0.3 mm			FD < 0.2 mm			FD < 0.1 mm		
	Impact	Over	Under	Impact	Over	Under	Impact	Over	Under	Impact	Over	Under
male	0	0	0	0	0.899	0	0	0.731	0	0.041	0.577	0.087
age_months	0	0	0.99	0.402	0.489	0.358	0.348	0.315	0.491	0.401	0.339	0.492
height	0.002	0.001	0.574	0	0	0.141	0.012	0	0.768	0.085	0.03	0.514
weight	0	0	0	0	0.131	0	0	0.243	0	0.022	0.35	0.116
bmi	0	0	0	0	0.386	0	0	0.828	0	0.108	0.769	0.108
home_roster	0.121	0.016	0.946	0.829	0.728	0.841	0.873	0.649	0.955	0.396	0.148	0.8
parental_income	0	1	0	0	1	0	0	1	0	0	0.99	0
parents_married	0	1	0	0	1	0	0	1	0	0	0.461	0.02
parental_education	0	1	0	0	1	0	0	1	0	0	0.779	0.001
residential_deprivation	0	0.999	0	0	1	0	0	1	0	0	0.996	0
mri_runs	0.004	0.007	0.063	0	0.012	0	0.014	0.129	0.151	0.083	0.679	0.029
sleepy	0.271	0.14	0.62	1	1	1	0.265	0.584	0.086	0.659	0.733	0.423
nihtbx_total	0	0.013	0.001	0.006	0.78	0	0.311	1	0.011	0.9	0.999	0.39
nihtbx_cryst	0	0.037	0	0	0.048	0	0.004	0.389	0	0.228	0.837	0.04
nihtbx_fluid	0.012	0.1	0.014	0.012	0.736	0.001	0.376	0.997	0.032	0.81	0.989	0.52
nihtbx_picvocab	0.039	0.71	0.013	0	0.296	0	0.024	0.591	0.006	0.328	0.799	0.096
nihtbx_flanker	0.068	0.149	0.106	0.021	0.329	0.004	0.203	0.745	0.072	0.901	0.977	0.756
nihtbx_list	0.815	0.97	0.745	0.877	1	0.483	0.709	0.979	0.451	0.949	0.994	0.775
nihtbx_cardsort	0.327	0.354	0.342	0.026	0.536	0.007	0.188	0.867	0.058	0.72	0.976	0.24
nihtbx_pattern	0.002	0.003	0.023	0.245	0.931	0.053	0.415	0.902	0.12	0.158	0.319	0.081
nihtbx_picture	0.003	0.002	0.056	0.048	0.216	0.032	0.598	0.832	0.459	0.895	0.989	0.74
nihtbx_reading	0	0	0	0.001	0.013	0.001	0.024	0.324	0.003	0.657	0.962	0.331
wiscv_matrix	0	0.898	0	0	0.999	0	0	1	0	0.272	0.985	0.033
cbcl_internal	0.321	0.137	0.797	0.095	0.056	0.184	0.547	0.544	0.721	0.491	0.514	0.467
cbcl_external	0.086	0.097	0.253	0.438	0.89	0.208	0.082	0.409	0.037	0.315	0.887	0.109
cbcl_total	0.013	0	0.537	0.048	0.108	0.071	0.265	0.267	0.252	0.368	0.704	0.143
cbcl_anxdep	0.16	0.058	0.436	0.002	0.001	0.006	0.1	0.11	0.147	0.372	0.381	0.379
cbcl_withdep	0.001	0.001	0.119	0.141	0.213	0.197	0.429	0.39	0.61	0.202	0.281	0.187
cbcl_somatic	0.976	0.874	0.999	0.055	0.038	0.119	0.753	0.646	0.841	0.621	0.556	0.631
cbcl_social	0.098	0.021	0.44	0	0.016	0.001	0.081	0.231	0.078	0.589	0.85	0.315
cbcl_thought	0.201	0.042	0.738	0.065	0.256	0.032	0.412	0.709	0.217	0.237	0.569	0.081
cbcl_attention	0.76	0.385	0.975	0.265	0.838	0.076	0.74	0.973	0.323	0.527	0.964	0.107
cbcl_rulebreak	0	0.06	0.001	0	0.25	0	0	0.247	0	0.138	0.805	0.027
cbcl_aggressive	0.113	0.115	0.167	0.001	0.124	0	0.062	0.439	0.021	0.108	0.799	0.019
pps_yes_num	0.001	0.001	0.993	0.048	0.288	0.321	0.033	0.107	0.552	0	0	0.388
pps_severity	0.15	0.035	0.973	0.365	0.804	0.237	0.371	0.552	0.475	0.009	0.016	0.212
bis	0.51	0.19	0.904	0.351	0.271	0.401	0.815	0.678	0.84	0.445	0.481	0.416
bas_rr	0.861	0.876	0.791	0.087	0.777	0.012	0.442	0.952	0.129	0.201	0.677	0.063
bas_drive	0.02	0.682	0.009	0.008	0.968	0.001	0.088	0.739	0.032	0.249	0.421	0.289
bas_fs	0.042	0.067	0.047	0.134	0.649	0.01	0.253	0.545	0.087	0.25	0.426	0.168
upps_perserverance	0.205	0.051	0.636	0.036	0.084	0.037	0.416	0.372	0.454	0.957	0.982	0.874
upps_planning	0.668	0.321	0.955	0.775	0.797	0.722	0.11	0.08	0.145	0.377	0.45	0.269
upps_seeking	0.247	0.225	0.518	0.035	0.085	0.062	0.316	0.559	0.219	0.871	0.983	0.681
upps_pos_urgency	0.381	0.775	0.321	1	1	0.644	0.167	0.802	0.127	0.827	0.9	0.719
upps_neg_urgency	0.053	0.01	0.74	0.013	0.038	0.023	0.046	0.089	0.089	0.555	0.688	0.525

Table S2: Significance (p-values) of motion impact score for selected variables in the ABCD study after motion-reduction with ABCD-BIDS and with (varying from left to right) no motion censoring, censoring at framewise displacement (FD) < 0.3 mm, FD < 0.2, and FD < 0.1. Omnibus p-values are given for overestimation scores, underestimation scores, and impact from both/either type of score. The omnibus p-values account for multiple comparisons across edges, but they are not corrected for the multiple comparisons across the 45 different traits shown.

Variable	SHAMAN Motion Score											
	No Censoring			FD < 0.3 mm			FD < 0.2 mm			FD < 0.1 mm		
	Impact	Over	Under	Impact	Over	Under	Impact	Over	Under	Impact	Over	Under
male	108.0	38.0	47.9	51.2	-8.2	45.9	48.8	-3.2	43.3	13.8	0.0	13.4
age_months	31.7	47.6	-12.7	3.0	2.2	4.0	4.1	4.7	2.1	3.0	4.0	1.7
height	24.3	29.9	1.5	28.7	29.4	8.2	17.9	24.8	-2.6	10.1	14.1	1.6
weight	100.2	56.9	45.5	62.1	13.6	40.5	53.9	7.8	37.6	15.5	4.9	11.7
bmi	147.3	91.9	53.7	70.4	5.6	52.2	64.7	-5.5	54.9	9.4	-3.5	11.5
home_roster	9.3	19.0	-6.7	-2.9	-1.7	-3.2	-4.2	-0.4	-7.7	3.4	9.4	-3.4
parental_income	128.0	-69.9	145.7	151.2	-80.5	165.0	106.6	-75.4	127.4	56.3	-21.7	62.7
parents_married	90.5	-36.6	98.5	82.9	-43.3	92.6	72.9	-31.3	76.0	35.4	2.7	26.7
parental_education	97.6	-55.7	115.4	110.6	-66.6	130.7	73.4	-49.9	92.0	33.1	-6.1	36.9
residential_deprivation	84.4	-19.6	86.3	82.0	-44.9	98.9	65.7	-44.2	84.8	39.7	-23.8	54.2
mri_runs	27.2	23.4	15.5	35.0	20.0	29.8	15.4	11.1	9.9	10.8	-1.5	16.4
sleepy	7.9	13.4	-1.0	-59.8	-61.7	-45.4	5.3	0.7	10.0	-0.6	-1.8	2.6
nihtbx_total	26.2	17.5	22.3	20.2	-2.4	31.8	4.5	-11.5	17.8	-5.2	-12.2	3.5
nihtbx_cryst	28.2	13.7	29.6	36.7	12.8	41.9	22.6	3.5	29.7	6.3	-3.4	13.3
nihtbx_fluid	16.0	9.4	15.0	16.5	-1.2	24.3	3.6	-9.9	12.4	-2.9	-8.9	1.8
nihtbx_picvocab	13.1	-1.0	18.7	25.7	5.0	32.6	15.5	0.2	23.6	4.5	-2.7	10.2
nihtbx_flanker	10.7	8.0	9.3	13.9	4.3	16.8	6.8	-1.3	10.7	-5.1	-7.6	-1.9
nihtbx_list	-1.8	-5.4	-0.5	-3.6	-11.5	1.8	-1.0	-7.4	2.5	-6.3	-10.2	-1.9
nihtbx_cardsort	4.7	4.2	4.6	13.4	1.2	19.2	7.0	-3.7	11.9	-1.4	-8.0	5.9
nihtbx_pattern	18.3	18.5	13.2	5.4	-4.7	11.7	3.2	-4.9	8.9	8.3	4.4	11.0
nihtbx_picture	18.9	20.1	12.0	11.2	5.9	12.4	0.2	-3.4	2.5	-5.2	-8.7	-1.7
nihtbx_reading	37.1	33.1	30.1	29.7	16.7	32.1	14.7	4.4	18.7	-0.4	-7.0	4.5
wiscv_matrix	45.8	-6.8	55.0	39.9	-18.3	53.7	26.7	-22.9	44.2	5.5	-12.1	16.9
cbcl_internal	4.6	8.3	-1.7	9.8	10.9	6.7	1.2	1.3	-1.1	1.9	1.7	2.2
cbcl_external	9.7	8.7	6.1	2.6	-5.7	7.5	9.9	3.0	12.4	4.5	-4.6	9.4
cbcl_total	14.8	22.0	2.0	11.2	9.1	10.5	5.3	5.3	5.4	3.6	-1.2	8.3
cbcl_anxdep	8.2	11.5	2.8	23.6	23.6	19.9	9.6	9.0	7.7	3.9	3.8	3.8
cbcl_withdep	25.2	29.8	9.3	8.1	6.3	6.9	2.5	3.2	0.2	7.1	5.5	7.7
cbcl_somatic	-7.1	-3.0	-9.9	11.5	12.6	8.4	-1.7	-0.2	-3.1	-0.1	0.9	-0.2
cbcl_social	11.0	17.1	2.9	24.1	15.5	24.8	9.7	6.0	10.4	0.4	-4.2	4.7
cbcl_thought	6.6	12.9	-0.9	10.8	5.5	13.3	2.9	-0.8	6.2	6.1	1.0	10.3
cbcl_attention	-1.0	3.9	-7.0	5.4	-3.8	10.8	-1.4	-8.3	4.8	1.5	-8.2	10.0
cbcl_rulebreak	25.9	11.6	29.0	32.8	6.6	40.5	26.2	6.7	31.3	8.7	-3.8	15.8
cbcl_aggressive	9.6	8.8	7.8	24.0	9.1	27.6	11.2	2.6	15.2	10.4	-3.0	18.0
pps_yes_num	19.6	34.8	-18.1	11.6	6.9	6.1	13.6	13.4	0.3	30.8	34.0	3.9
pps_severity	8.5	19.5	-12.9	3.9	-4.5	8.0	3.7	0.5	2.2	20.9	20.9	8.6
bis	2.2	6.8	-3.6	3.7	5.0	3.1	-2.8	-0.6	-3.0	2.5	2.2	3.0
bas_rr	-2.7	-3.1	-1.7	10.2	-2.2	17.5	2.5	-6.6	8.9	6.9	-0.9	11.3
bas_drive	15.2	-0.9	21.6	17.1	-10.2	29.7	10.2	-3.0	17.5	5.8	3.2	5.8
bas_fs	11.6	10.2	11.6	8.2	-0.2	15.0	5.7	1.0	10.5	5.9	2.9	7.4
upps_perserverance	7.4	11.4	0.4	12.7	9.9	12.0	2.8	3.4	2.3	-6.7	-8.3	-4.1
upps_planning	0.3	4.5	-5.3	-1.9	-2.2	-1.0	9.3	10.0	7.9	3.6	2.6	5.6
upps_seeking	5.8	6.0	2.0	12.6	9.6	11.3	4.4	1.1	5.9	-4.1	-8.1	-0.8
upps_pos_urgency	4.1	-2.7	5.1	-18.6	-32.7	-1.7	7.1	-3.7	10.1	-3.1	-6.2	-2.0
upps_neg_urgency	11.7	17.4	-0.7	16.3	13.5	14.7	11.6	9.8	10.1	0.7	-1.2	1.2

Table S3: Motion impact scores (omnibus Stouffer's Z) for selected variables in the ABCD study after motion-reduction with ABCD-BIDS and with (varying from left to right) no motion censoring, censoring at framewise displacement (FD) < 0.3 mm, FD < 0.2, and FD < 0.1. Higher scores correspond to a greater impact of motion. Motion overestimation scores, underestimation scores, and overall impact (both/either over/underestimation) are shown separately.

Skew Table

Variable	Percent Difference in Mean Compared to No Censoring		
	FD < 0.3 mm	FD < 0.2 mm	FD < 0.1 mm
male	-0.93%	-1.98%	-4.19%
age_months	0.53%	0.98%	2.43%
height	0.12%	0.15%	0.13%
weight	-0.09%	-0.23%	-0.97%
bmi	-0.15%	-0.34%	-1.33%
home_roster	0.04%	0.15%	0.35%
parental_income	0.29%	0.64%	1.94%
parents_married	0.30%	0.77%	2.87%
parental_education	0.15%	0.24%	0.84%
residential_deprivation	-0.28%	-0.44%	-1.52%
mri_runs	0.88%	1.25%	1.95%
sleepy	-0.11%	-0.19%	-0.73%
nihtbx_total	0.17%	0.33%	0.80%
nihtbx_cryst	0.10%	0.18%	0.45%
nihtbx_fluid	0.16%	0.35%	0.81%
nihtbx_picvocab	0.07%	0.11%	0.29%
nihtbx_flanker	0.05%	0.12%	0.29%
nihtbx_list	0.11%	0.21%	0.44%
nihtbx_cardsort	0.14%	0.31%	0.64%
nihtbx_pattern	0.15%	0.32%	0.81%
nihtbx_picture	0.10%	0.21%	0.55%
nihtbx_reading	0.09%	0.19%	0.45%
wiscv_matrix	0.22%	0.55%	1.20%
cbcl_internal	-0.12%	-0.22%	-0.37%
cbcl_external	-0.37%	-0.51%	-1.03%
cbcl_total	-0.31%	-0.51%	-1.10%
cbcl_anxdep	-0.06%	-0.13%	-0.07%
cbcl_withdep	-0.16%	-0.29%	-0.51%
cbcl_somatic	0.01%	0.03%	-0.03%
cbcl_social	-0.18%	-0.34%	-0.71%
cbcl_thought	-0.20%	-0.33%	-0.67%
cbcl_attention	-0.24%	-0.40%	-0.88%
cbcl_rulebreak	-0.24%	-0.37%	-0.65%
cbcl_aggressive	-0.17%	-0.28%	-0.59%
pps_yes_num	-0.25%	-0.37%	-0.86%
pps_severity	-0.15%	-0.21%	-0.47%
bis	0.00%	-0.07%	-0.14%
bas_rr	-0.10%	-0.19%	-0.59%
bas_drive	-0.35%	-0.63%	-0.92%
bas_fs	-0.18%	-0.33%	-0.59%
upps_perserverance	-0.24%	-0.38%	-0.74%
upps_planning	-0.32%	-0.38%	-0.68%
upps_seeking	-0.02%	0.04%	0.25%
upps_pos_urgency	-0.33%	-0.37%	-1.08%
upps_neg_urgency	-0.30%	-0.43%	-0.92%

Table S4: Sampling bias at different levels of motion censoring. The percentage difference in sample mean (compared to no censoring) is shown for selected variables in ABCD after censoring at a framewise displacement (FD) of 0.3, 0.2, or < 0.1 mm. Two variables (gender and number of MRI runs completed) are biased by more than 1% at FD < 0.2 mm, and 11 variables are biased by > 1% at FD < 0.1 mm.

Figure S4: Sampling bias at different levels of framewise censoring for selected variables. Sample distributions are shown for (a) body mass index (BMI), (b) WISC-V matrix reasoning score, and (c) sex assigned at birth. Mean (solid line) and standard deviation (SD) for (a) BMI and (b) matrix reasoning were similar across all levels of motion censoring, but the proportion of boys to girls shifted significantly from 52.6% without motion censoring to 51.6% with very stringent censoring at framewise displacement (FD) < 0.1 mm.

Figure S5: Sampling bias at different levels of framewise censoring. Box and whisker plots are shown for all 45 traits examined in the ABCD study at different levels of motion censoring. From left to right, no censoring, FD < 0.3 mm, FD < 0.2 mm, and FD < 0.1 mm.

Figure SB.2: Effects of frame censoring on motion impact score in HCP data using FD. Motion impact score (omnibus Stouffer's Z, higher = more motion) for traits of a given category at different levels of motion censoring. The cutoff score for significance at $p < 0.05$ is different for each trait. The range of significance thresholds is indicated by a gray band. **(a)** Motion overestimation score. **(b)** Motion underestimation score.

Higher scores correspond to a greater impact of motion. Motion overestimation scores, underestimation scores, and overall impact (both/either over/underestimation) are shown separately.

Two brief additional comments regarding the DVARS analysis: First, I couldn't find many details about the DVARS implementation, but based on Fig. SB.3 it appears that a cutoff of 200, rather than the statistically principled dual cutoff Afyouni and Nichols (2018), was adopted. Since that principled cutoff technique was found to be superior, it would be preferable to adopt it. Second, the amount of volumes removed via DVARS vs. FD is not reported, it seems, which is relevant information when comparing scrubbing methods.

We have updated our DVARS analysis to use the “standardized DVARS” methods proposed by Afyouni and Nichols in the Neuroimage 2018 paper suggested by the reviewer. We use the publicly-available implementation in the `nipype.algorithms.confounds.compute_dvars` method, part of the Nipype Neuroimaging in Python toolkit. The output is normalized by 1000 (that is, DVARS 1.2 → 200) so that it can be reported on the same scale as conventional DVARS.

Also, as suggested by the reviewer, we now report the number of volumes removed for both DVARS and FD.

Motion Impact Scores Were Similar for FD and DVARS in ABCD

Motion impact scores, and the effect of motion censoring, were comparable for FD and DVARS. There were 7270 participants with usable fMRI in ABCD and 1017 participants in HCP prior to censoring. Frame censoring at FD < 0.3 mm retained 6886 participants in ABCD and 1017 in HCP, 0.2 mm left 6452 in ABCD and 1017 in HCP, and 0.1 mm left 4558 in ABCD and 928 in HCP. Frame censoring at DVARS < 200 retained 6212 participants in ABCD and 1017 in HCP.

Figure SB.4: Effects of frame censoring on motion impact score in HCP data using DVARS. Motion impact score (omnibus Stouffer's Z, higher = more motion) for traits of a given category at different levels of motion censoring. The cutoff score for significance at $p < 0.05$ is different for each trait. The range of significance thresholds is indicated by a gray band. **(a)** Motion overestimation score. **(b)** Motion underestimation score.

Variable	Correlation w/DVARS		SHAMAN p-Value			Shaman Motion Score		
	Pearson	Spearman	Impact	Over	Under	Impact	Over	Under
Age_in_Yrs	-0.189	-0.179	0.969	0.969	0.961	-24.00	-23.69	-20.91
Height	0.36	0.361	0.078	0.375	0.039	66.71	20.44	65.81
Weight	0.11	0.144	0	0.102	0.008	95.02	47.93	66.56
BMI	-0.101	-0.055	0.391	0.352	0.805	19.54	24.58	0.88
Handedness	-0.033	-0.052	0.039	0.039	0.039	103.99	95.65	99.59
SSAGA_Income	0.045	0.047	0.328	0.352	0.328	24.01	21.05	24.04
SSAGA_Educ	0.088	0.053	0.398	0.734	0.273	20.20	1.64	29.01
Hematocrit_1	0.26	0.27	0.281	0.336	0.281	29.85	25.33	30.22
ThyroidHormone	0.025	0.013	0.063	0.078	0.063	96.90	89.11	96.48
HbA1C	-0.124	-0.199	0.016	0.016	0.016	126.72	119.91	123.84
MMSE_Score	0.008	-0.011	0.82	0.82	0.836	-11.41	-10.43	-11.57
PSQI_Score	-0.078	-0.058	0.969	0.969	0.977	-53.49	-49.73	-50.02
PicSeq_AgeAdj	-0.052	-0.06	0.109	0.086	0.094	52.55	52.42	49.80
CardSort_AgeAdj	-0.025	-0.024	0.977	0.977	0.984	-28.40	-25.49	-29.66
Flanker_AgeAdj	-0.033	-0.028	0.875	0.914	0.867	-14.03	-14.51	-9.26
PMAT24_A_CR	0.149	0.138	0.453	0.383	0.5	14.27	18.08	8.18
PMAT24_A_RTCT	0.112	0.094	0.086	0.898	0.188	68.93	-49.78	61.68
ReadEng_AgeAdj	0.09	0.075	0.75	0.734	0.719	-2.16	-1.70	1.68
PicVocab_AgeAdj	0.133	0.123	0.234	0.344	0.172	35.18	23.73	37.93
ProcSpeed_AgeAdj	-0.049	-0.033	0.227	0.281	0.227	36.77	31.76	37.67
DDisc_AUC_200	0.107	0.087	0.453	0.438	0.484	16.55	17.17	15.57
DDisc_AUC_40K	0.132	0.132	0.984	0.977	1	-29.89	-24.10	-30.55
VSPLOT_TC	0.109	0.099	0.898	0.844	0.898	-12.15	-7.66	-11.18
VSPLOT_CRTE	-0.04	-0.07	0.977	0.984	0.969	-77.52	-75.46	-71.59
IWRD_TOT	-0.053	-0.048	0.727	0.781	0.766	-4.79	-5.59	-2.79
IWRD_RTC	-0.028	-0.029	0.148	0.156	0.148	53.74	52.01	52.66
ListSort_AgeAdj	0.038	0.042	0.914	0.922	0.875	-19.07	-19.84	-15.14
CogFluidComp_AgeAdj	-0.037	-0.047	0.703	0.664	0.727	0.86	4.04	-0.79
CogTotalComp_AgeAdj	0.05	0.035	0.336	0.32	0.383	25.34	27.11	19.00
CogCrystalComp_AgeAdj	0.122	0.1	0.742	0.781	0.68	-2.37	-3.16	2.68
ER40_CR	-0.052	-0.036	0.586	0.531	0.688	8.84	12.53	-0.30
ER40_CRT	0.025	-0.028	0.32	0.328	0.305	33.56	29.38	30.37
AngAffect_Unadj	0.037	0.062	0.313	0.32	0.367	25.23	25.31	22.09
AngHostil_Unadj	-0.01	-0.015	0.477	0.438	0.594	15.86	18.01	9.77
AngAggr_Unadj	0.031	0.039	0.172	0.32	0.148	42.71	24.68	44.18
FearAffect_Unadj	-0.041	-0.039	0.898	0.891	0.914	-27.31	-22.81	-30.15
FearSomat_Unadj	-0.028	-0.032	0.672	0.766	0.57	5.59	-0.31	10.84
Sadness_Unadj	-0.052	-0.046	0.875	0.875	0.883	-29.41	-26.80	-29.16
LifeSatisf_Unadj	0.047	0.04	0.109	0.117	0.109	54.59	52.87	53.76
MeanPurp_Unadj	-0.019	-0.014	0.922	0.914	0.922	-19.19	-18.57	-17.76
PosAffect_Unadj	-0.009	0.007	0.328	0.383	0.297	27.71	24.24	29.50
Friendship_Unadj	0.072	0.045	0.07	0.063	0.102	65.39	65.84	55.39
Loneliness_Unadj	-0.024	-0.036	0.336	0.336	0.328	25.13	24.51	25.04
PercHostil_Unadj	0.011	0.015	0.422	0.445	0.461	23.00	21.42	20.45
PercReject_Unadj	-0.078	-0.07	0.461	0.492	0.445	12.89	11.01	14.02
EmotSupp_Unadj	0.067	0.04	0.023	0.016	0.023	109.18	106.08	104.48
InstruSupp_Unadj	0.033	0.024	0.391	0.398	0.391	21.13	20.16	21.24
PercStress_Unadj	-0.057	-0.053	0.063	0.07	0.055	61.73	54.65	63.93
SelfEff_Unadj	0.045	0.054	0.359	0.422	0.352	24.43	20.19	25.38
Dexterity_AgeAdj	-0.128	-0.117	0.07	0.039	0.102	72.10	73.74	56.86
Strength_AgeAdj	0.304	0.318	0.766	0.977	0.68	-4.02	-18.45	6.29
NEOFAC_A	-0.068	-0.066	0.156	0.203	0.164	45.37	37.09	43.33
NEOFAC_O	0.125	0.095	0.992	0.992	0.992	-49.82	-52.10	-43.31
NEOFAC_C	-0.051	-0.053	0.953	0.953	0.984	-37.23	-31.34	-33.38
NEOFAC_N	-0.06	-0.058	1	1	0.992	-41.67	-40.28	-34.37
NEOFAC_E	0.085	0.093	0.156	0.195	0.164	41.17	39.01	41.35
Noise_Comp	-0.091	-0.102	0.078	0.078	0.094	65.94	61.11	58.08
Odor_AgeAdj	-0.082	-0.127	1	1	1	-43.06	-38.84	-42.64
Taste_AgeAdj	-0.074	-0.063	0.063	0.055	0.063	70.95	68.53	65.78
ASR_Anxd_Pct	-0.001	0.015	0.938	0.922	0.914	-41.77	-40.49	-34.03
ASR_Witd_T	-0.019	0.012	0.102	0.094	0.094	60.70	57.89	60.36
ASR_Soma_T	-0.091	-0.023	0.891	0.883	0.898	-20.15	-17.10	-19.28
ASR_Thot_T	-0.022	-0.054	0.641	0.695	0.633	2.21	-0.06	4.38
ASR_Attn_T	-0.01	0.042	1	0.984	0.992	-88.00	-80.38	-76.59
ASR_Aggr_T	-0.014	-0.015	0.789	0.805	0.789	-13.27	-13.00	-12.31
ASR_Rule_T	-0.034	-0.064	0.977	0.984	0.953	-35.09	-37.87	-30.56
ASR_Intr_T	0.021	0.042	0.758	0.813	0.766	-10.23	-13.52	-5.57
ASR_Intn_T	-0.023	-0.018	0.844	0.852	0.836	-13.10	-10.73	-11.03
ASR_Extn_T	0.001	0.013	0.5	0.648	0.469	16.71	6.96	17.05
ASR_Totp_T	-0.008	0.002	0.992	0.992	0.992	-41.67	-37.08	-40.97
DSM_Depr_T	-0.058	-0.071	0.828	0.852	0.828	-19.39	-19.74	-14.65
DSM_Anxi_T	-0.02	-0.024	0.984	0.992	0.984	-86.50	-82.91	-79.41
DSM_Somp_T	-0.088	-0.075	0.859	0.852	0.875	-14.84	-12.99	-13.00
DSM_Avoid_T	0.01	0.056	0.352	0.383	0.359	25.13	20.94	24.47
DSM_Adh_T	0.019	0.083	1	1	1	-92.33	-84.95	-82.90
DSM_Antis_T	-0.045	-0.09	0.984	0.992	0.977	-51.62	-52.12	-44.78

Table SB.7: Relationship between selected variables in the HCP study with head motion (DVARS) averaged over each participant's resting state fMRI scans. The SHAMAN (omnibus) p-values and motion scores (omnibus Stouffer's Z-scores) are also given for overestimation scores, underestimation scores, and impact from both/either type of score. Results in this table reflect data after motion-reduction and without additional motion censoring.

Variable	SHAMAN p-Value					
	No Censoring			DVARS < 200		
	Impact	Over	Under	Impact	Over	Under
Age_in_Yrs	0.969	0.969	0.961	0.883	0.891	0.883
Height	0.078	0.375	0.039	0.063	0.273	0.047
Weight	0	0.102	0.008	0.078	0.313	0.148
BMI	0.391	0.352	0.805	0.789	0.703	0.914
Handedness	0.039	0.039	0.039	0.016	0.023	0.016
SSAGA_Income	0.328	0.352	0.328	0.492	0.547	0.5
SSAGA_Educ	0.398	0.734	0.273	0.961	0.992	0.969
Hematocrit_1	0.281	0.336	0.281	0.227	0.242	0.234
ThyroidHormone	0.063	0.078	0.063	0.039	0.055	0.039
HbA1C	0.016	0.016	0.016	0	0	0
MMSE_Score	0.82	0.82	0.836	0.734	0.719	0.75
PSQL_Score	0.969	0.969	0.977	0.984	0.969	0.984
PicSeq_AgeAdj	0.109	0.086	0.094	0.313	0.289	0.32
CardSort_AgeAdj	0.977	0.977	0.984	0.93	0.898	0.953
Flanker_AgeAdj	0.875	0.914	0.867	0.922	0.938	0.891
PMAT24_A_CR	0.453	0.383	0.5	0.516	0.461	0.68
PMAT24_A_RTCT	0.086	0.898	0.188	0.031	0.984	0.063
ReadEng_AgeAdj	0.75	0.734	0.719	0.914	0.797	0.922
PicVocab_AgeAdj	0.234	0.344	0.172	0.844	0.891	0.828
ProcSpeed_AgeAdj	0.227	0.281	0.227	0.219	0.258	0.195
DDisc_AUC_200	0.453	0.438	0.484	0.688	0.625	0.711
DDisc_AUC_40K	0.984	0.977	1	1	0.992	1
VSPLOT_TC	0.898	0.844	0.898	0.914	0.891	0.938
VSPLOT_CRTE	0.977	0.984	0.969	0.961	0.969	0.961
IWRD_TOT	0.727	0.781	0.766	0.469	0.484	0.523
IWRD_RTC	0.148	0.156	0.148	0.086	0.102	0.078
ListSort_AgeAdj	0.914	0.922	0.875	0.852	0.891	0.844
CogFluidComp_AgeAdj	0.703	0.664	0.727	0.789	0.727	0.789
CogTotalComp_AgeAdj	0.336	0.32	0.383	0.633	0.547	0.766
ogCrystalComp_AgeAdj	0.742	0.781	0.68	0.945	0.891	0.953
ER40_CR	0.586	0.531	0.688	0.688	0.648	0.781
ER40_CRT	0.32	0.328	0.305	0.25	0.266	0.273
AngAffect_Unadj	0.313	0.32	0.367	0.875	0.883	0.891
AngHostil_Unadj	0.477	0.438	0.594	0.5	0.484	0.57
AngAggr_Unadj	0.172	0.32	0.148	0.633	0.789	0.648
FearAffect_Unadj	0.898	0.891	0.914	0.969	0.961	0.969
FearSomat_Unadj	0.672	0.766	0.57	0.719	0.797	0.672
Sadness_Unadj	0.875	0.875	0.883	0.953	0.953	0.961
LifeSatisf_Unadj	0.109	0.117	0.109	0.211	0.234	0.195
MeanPurp_Unadj	0.922	0.914	0.922	0.891	0.891	0.883
PosAffect_Unadj	0.328	0.383	0.297	0.313	0.359	0.289
Friendship_Unadj	0.07	0.063	0.102	0.086	0.078	0.125
Loneliness_Unadj	0.336	0.336	0.328	0.555	0.555	0.547
PercHostil_Unadj	0.422	0.445	0.461	0.742	0.75	0.773
PercReject_Unadj	0.461	0.492	0.445	0.883	0.898	0.859
EmotSupp_Unadj	0.023	0.016	0.023	0.07	0.07	0.07
InstruSupp_Unadj	0.391	0.398	0.391	0.531	0.539	0.531
PercStress_Unadj	0.063	0.07	0.055	0.164	0.18	0.148
SelfEif_Unadj	0.359	0.422	0.352	0.344	0.375	0.328
Dexterity_AgeAdj	0.07	0.039	0.102	0.188	0.164	0.281
Strength_AgeAdj	0.766	0.977	0.68	0.828	0.953	0.828
NEOFAC_A	0.156	0.203	0.164	0.305	0.336	0.336
NEOFAC_O	0.992	0.992	0.992	0.984	1	0.992
NEOFAC_C	0.953	0.953	0.984	0.984	0.977	0.977
NEOFAC_N	1	1	0.992	0.945	0.961	0.938
NEOFAC_E	0.156	0.195	0.164	0.195	0.227	0.203
Noise_Comp	0.078	0.078	0.094	0.094	0.102	0.133
Odor_AgeAdj	1	1	1	0.992	0.992	1
Taste_AgeAdj	0.063	0.055	0.063	0.164	0.164	0.203
ASR_Anxd_Pct	0.938	0.922	0.914	0.93	0.938	0.914
ASR_Witd_T	0.102	0.094	0.094	0.164	0.164	0.156
ASR_Soma_T	0.891	0.883	0.898	0.836	0.836	0.859
ASR_Thot_T	0.641	0.695	0.633	0.82	0.844	0.828
ASR_Attn_T	1	0.984	0.992	0.992	0.992	0.992
ASR_Aggr_T	0.789	0.805	0.789	0.922	0.938	0.93
ASR_Rule_T	0.977	0.984	0.953	0.969	0.969	0.953
ASR_Intr_T	0.758	0.813	0.766	0.797	0.883	0.773
ASR_Intrn_T	0.844	0.852	0.836	0.836	0.836	0.844
ASR_Extn_T	0.5	0.648	0.469	0.758	0.852	0.664
ASR_Totp_T	0.992	0.992	0.992	1	1	1
DSM_Depr_T	0.828	0.852	0.828	0.82	0.852	0.813
DSM_Anxi_T	0.984	0.992	0.984	1	1	1
DSM_Somp_T	0.859	0.852	0.875	0.93	0.922	0.93
DSM_Avoid_T	0.352	0.383	0.359	0.508	0.484	0.508
DSM_Adh_T	1	1	1	1	1	1
DSM_Antis_T	0.984	0.992	0.977	0.992	0.992	0.992

Table SB.8: Significance (p-values) of motion impact score for selected variables in the HCP study after motion-reduction and with (varying from left to right) no motion censoring and censoring at DVARS < 200. Omnibus p-values are given for overestimation scores, underestimation scores, and impact from both/either type of score. The omnibus p-values account for multiple comparisons across edges, but they are not corrected for the multiple comparisons across the 76 different traits shown.

Variable	SHAMAN Motion Score					
	No Censoring			DVARs < 200		
	Impact	Over	Under	Impact	Over	Under
Age_in_Yrs	-24.0	-23.7	-20.9	-11.2	-12.0	-7.8
Height	66.7	20.4	65.8	73.4	32.8	63.1
Weight	95.0	47.9	66.6	59.2	28.2	42.7
BMI	19.5	24.6	0.9	-6.9	2.6	-15.4
Handedness	104.0	95.7	99.6	102.9	94.8	98.2
SSAGA_Income	24.0	21.0	24.0	14.0	11.0	14.2
SSAGA_Educ	20.2	1.6	29.0	-28.5	-29.3	-21.7
Hematocrit_1	29.8	25.3	30.2	41.9	36.3	38.8
ThyroidHormone	96.9	89.1	96.5	102.2	94.1	101.4
HbA1C	126.7	119.9	123.8	136.9	130.2	133.7
MMSE_Score	-11.4	-10.4	-11.6	-4.7	-4.4	-5.3
PSQL_Score	-53.5	-49.7	-50.0	-52.2	-46.1	-51.0
PicSeq_AgeAdj	52.6	52.4	49.8	24.6	26.7	22.4
CardSort_AgeAdj	-28.4	-25.5	-29.7	-17.9	-15.4	-18.7
Flanker_AgeAdj	-14.0	-14.5	-9.3	-20.0	-19.3	-14.5
PMAT24_A_CR	14.3	18.1	8.2	9.5	14.2	2.4
PMAT24_A_RTCT	68.9	-49.8	61.7	103.3	-95.3	116.9
ReadEng_AgeAdj	-2.2	-1.7	1.7	-12.5	-4.3	-13.3
PicVocab_AgeAdj	35.2	23.7	37.9	-12.0	-12.4	-7.3
ProcSpeed_AgeAdj	36.8	31.8	37.7	40.7	36.3	40.6
DDisc_AUC_200	16.5	17.2	15.6	0.4	3.4	-0.3
DDisc_AUC_40K	-29.9	-24.1	-30.5	-27.1	-19.8	-29.6
VSPLOT_TC	-12.1	-7.7	-11.2	-21.1	-14.2	-22.0
VSPLOT_CRTE	-77.5	-75.5	-71.6	-68.1	-66.6	-63.0
IWRD_TOT	-4.8	-5.6	-2.8	14.1	11.2	12.0
IWRD_RTC	53.7	52.0	52.7	56.9	54.2	56.2
ListSort_AgeAdj	-19.1	-19.8	-15.1	-17.7	-18.2	-14.3
CogFluidComp_AgeAdj	0.9	4.0	-0.8	-2.5	0.9	-3.7
CogTotalComp_AgeAdj	25.3	27.1	19.0	4.1	10.9	-1.2
CogCrystalComp_AgeAdj	-2.4	-3.2	2.7	-25.6	-14.2	-22.1
ER40_CR	8.8	12.5	-0.3	-0.2	3.7	-7.1
ER40_CRT	33.6	29.4	30.4	36.9	34.2	32.6
AngAffect_Unadj	25.2	25.3	22.1	-15.8	-16.1	-16.1
AngHostil_Unadj	15.9	18.0	9.8	14.4	16.4	8.9
AngAggr_Unadj	42.7	24.7	44.2	7.5	0.1	7.2
FearAffect_Unadj	-27.3	-22.8	-30.1	-37.8	-35.1	-37.8
FearSomat_Unadj	5.6	-0.3	10.8	0.6	-2.9	5.3
Sadness_Unadj	-29.4	-26.8	-29.2	-39.6	-38.1	-37.9
LifeSatisf_Unadj	54.6	52.9	53.8	37.9	36.6	38.0
MeanPurp_Unadj	-19.2	-18.6	-17.8	-19.0	-18.2	-17.2
PosAffect_Unadj	27.7	24.2	29.5	26.3	24.5	27.5
Friendship_Unadj	65.4	65.8	55.4	62.2	63.5	50.3
Loneliness_Unadj	25.1	24.5	25.0	8.0	9.6	9.4
PercHostil_Unadj	23.0	21.4	20.5	-6.0	-7.0	-7.5
PercReject_Unadj	12.9	11.0	14.0	-21.7	-23.1	-18.9
EmotSupp_Unadj	109.2	106.1	104.5	78.7	76.7	76.2
InstruSupp_Unadj	21.1	20.2	21.2	10.3	10.3	10.5
PercStress_Unadj	61.7	54.6	63.9	46.2	40.7	49.0
SelfEif_Unadj	24.4	20.2	25.4	27.2	23.5	27.6
Dexterity_AgeAdj	72.1	73.7	56.9	42.6	44.6	31.5
Strength_AgeAdj	-4.0	-18.4	6.3	-8.3	-13.0	-4.0
NEOFAC_A	45.4	37.1	43.3	28.1	24.4	24.7
NEOFAC_O	-49.8	-52.1	-43.3	-47.5	-47.4	-43.7
NEOFAC_C	-37.2	-31.3	-33.4	-45.6	-37.0	-42.2
NEOFAC_N	-41.7	-40.3	-34.4	-43.7	-41.1	-37.4
NEOFAC_E	41.2	39.0	41.3	36.2	34.7	36.1
Noise_Comp	65.9	61.1	58.1	57.7	53.7	50.3
Odor_AgeAdj	-43.1	-38.8	-42.6	-42.6	-38.0	-41.8
Taste_AgeAdj	70.9	68.5	65.8	53.3	51.3	48.1
ASR_Anxd_Pct	-41.8	-40.5	-34.0	-54.7	-50.7	-47.3
ASR_Witd_T	60.7	57.9	60.4	55.3	53.7	54.0
ASR_Soma_T	-20.1	-17.1	-19.3	-17.5	-13.4	-18.1
ASR_Thot_T	2.2	-0.1	4.4	-9.5	-10.1	-8.1
ASR_Attn_T	-88.0	-80.4	-76.6	-97.2	-88.3	-85.4
ASR_Aggr_T	-13.3	-13.0	-12.3	-36.8	-35.8	-34.6
ASR_Rule_T	-35.1	-37.9	-30.6	-45.0	-46.7	-39.0
ASR_Intr_T	-10.2	-13.5	-5.6	-9.6	-13.9	-5.2
ASR_Intrn_T	-13.1	-10.7	-11.0	-18.3	-15.1	-16.2
ASR_Extn_T	16.7	7.0	17.1	-9.1	-17.3	-1.3
ASR_Totp_T	-41.7	-37.1	-41.0	-51.2	-45.3	-50.0
DSM_Depr_T	-19.4	-19.7	-14.7	-33.9	-32.7	-30.1
DSM_Anxi_T	-86.5	-82.9	-79.4	-77.2	-73.2	-71.5
DSM_Somp_T	-14.8	-13.0	-13.0	-23.3	-21.3	-21.7
DSM_Avoid_T	25.1	20.9	24.5	15.1	14.5	13.2
DSM_Adh_T	-92.3	-84.9	-82.9	-100.1	-90.7	-90.3
DSM_Antis_T	-51.6	-52.1	-44.8	-66.7	-67.0	-59.2

Table SB.9: Motion impact scores (omnibus Stouffer's Z) for selected variables in the HCP study after motion-reduction and with (varying from left to right) no motion censoring and censoring at DVAR < 200. Higher scores correspond to a greater impact of motion. Motion overestimation scores, underestimation scores, and overall impact (both/either over/underestimation) are shown separately.

Minor Issues

Finally, the authors partially addressed another major concern from my earlier review:

3. “The use of statistical methods that assume linearity without verification”: The authors now report Spearman rank correlations, which do not depend on an assumption of linearity. However, I find it problematic that Pearson correlations are still reported, which is not appropriate when the data clearly exhibit non-linear associations.

We have now removed many mentions of Pearson correlation for measures for which it was not relevant.

On a related point, it is clear from some of the plots that some measures exhibit high degrees of right skew and should probably be log-transformed prior to fitting lines through them. For example, both x- and y-variables in Fig. S1a, and the y-variable in Fig. S2.

We revised Figure S1 to show a log-log plot of head motion vs fMRI variance. We no longer report the coefficient of determination (related to Pearson correlation). Instead, we approximate percentage of variance explained as the square of Spearman ρ . Using these new methods, our Results have been revised to report that ABCD-BIDS produces a 69% relative reduction in fMRI signal variance related to head motion.

The Effect of Residual Motion is Large Even After Denoising and Censoring

In order to characterize the impact of residual head motion on trait-FC effects, we first performed preliminary analyses to quantify how much residual motion was left in the data after denoising. Of the 11,874 children recruited into ABCD, $n = 9,652$ children with at least 8 minutes of rs-fMRI data were included in this portion of the analysis. ABCD-BIDS is the default denoising algorithm for pre-processed ABCD data^{25,37}. It includes global signal regression, respiratory filtering, spectral (low-pass) filtering, despiking, and regressing out the motion parameter timeseries. The relative performance of ABCD-BIDS was evaluated by comparing how much of the between-participant variability in the fMRI timeseries (averaged across regions of interest) was explained by head motion (framewise displacement, FD) in a linear, **log-log transformed** model before and after applying ABCD-BIDS (Figure S1). After minimal processing³⁹ (i.e. motion-correction by frame realignment only), **73%** of signal variance **(as estimated by taking the square of Spearman’s rho)** was explained by head motion. After further denoising using ABCD-BIDS (respiratory filtering, motion timeseries regression, despiking/interpolation of high-motion frames), **23%** of signal variance was explained by head motion. Therefore, ABCD-BIDS achieved a relative reduction in the proportion of signal variance related to motion of **69%** compared to minimal processing alone (see Methods).

fMRI BOLD Signal Variance Explained by Motion Before/After Processing

Figure S1: Proportion of fMRI BOLD signal variance explained by head motion before and after processing with the ABCD-BIDS (DCAN Labs) motion reduction algorithm. The comparison is made prior to any framewise motion censoring. **(a)** Log-log plot of variance of the mean node timeseries vs. mean framewise displacement (FD, in mm) for each child ($n = 9,652$). The square of Spearman's rho, ρ^2 is provided for the log-log best fit. **(b)** Proportion of variance of mean node (parcel) timeseries explained by motion (Pearson's rho squared, log-log fit) visualized on the cortical surface before and **(c)** after ABCD-BIDS.

Figure S2 was revised to show semi-log plots.

Figure S2: Correlation between trait variables and in-scanner head motion. (a) Body mass index (BMI, kg/m², mean 16.8, standard deviation 3.1), Spearman $\rho = 0.13$. **(b)** Wechsler Intelligence Scale for Children 5th Edition (WISC-V)⁵¹ matrix reasoning score (mean 10, standard deviation 3, maximum 19), Spearman $\rho = -0.12$. Both correlations were significant at $p < 0.001$. Each point represents one participant in the ABCD study. Head motion was measured as framewise displacement (FD, in mm) averaged over resting-state scans. FD was log-transformed to improve fit. The best fit is shown as a dashed line.

I wish the authors well in this work, and I hope they find these perspectives helpful.

Thank you very much for taking the time to help further improve this manuscript!

Reviewer #3

The resubmission has been substantially responsive to prior reviews and is substantively improved. However, there are two areas which were identified by Reviewer #1 in the first reviews which still require addressing:

1. "Our axiomatic assumption is that any signal that would affect inference is an "artifact" in the statistical sense, even if that signal arises from a real neurobiological process such as motor cortex activation, FC state, etc." *I disagree that this should be called an "artifact", a term that implies it is not a real biological effect in common usage. The authors need to acknowledge this is a short-coming of the proposed method (that real connectivity differences could be tied to motion increases/decreases) and avoid calling it an artifact.*

We appreciate this perspective from both Reviewer #1 and #3. As noted in the response to Reviewer #1, we have revised our discussion to clarify the distinction between artifact and biologically-driven fMRI signal changes.

Motion Impact May Arise from Heterogeneous Sources

The SHAMAN method distinguishes between transient (state) and persistent (trait) changes in fMRI signal related to a nuisance timeseries (head motion), but it does not distinguish between different biophysical sources of transient fMRI signal. Such sources could include not only artifactual signal from proton spin effects, but also fMRI activation in motor cortex, and even the transition between different FC states⁵⁶. Researchers specifically studying the relationship between small head movements and transient changes in neuronal activity might employ our SHAMAN method to discover interesting FC changes related to transient head motion. For more conventional BWAS, any fMRI signal change time-locked with head motion, regardless of source, has the potential to affect reproducibility if not removed or controlled for.

Motion May Negatively Affect the Signal-to-Noise-Ratio (SNR)

...the motion-FC effect arises from a combination of artifact, signal loss, and biologically-meaningful motion-related traits and FC states.

2. Linearity is still assumed in most models (Pearson R, linear correlation, simulations). There is no need to assume this, as there are plenty of non-linear regression methods that could be employed (e.g., random forests).

It is certainly true that linear assumptions are not appropriate in all models. We have relaxed linear assumptions implied by Pearson correlation by instead using Spearman correlation where appropriate, as also suggested by Reviewer #1.

Response to Reviewers

We appreciate the insightful feedback from Reviewer #1. The reviewer's comments were straightforward to address. We are grateful for the opportunity to revise and further strengthen the manuscript.

The reviewer's comments are highlighted gray. Our comments are in blue text, and excerpts from the manuscript are indented with changes highlighted in yellow.

Reviewer #1

Remarks to the Author

I appreciate the effort the authors have made to respond to my earlier feedback. The Discussion section especially is now more nuanced and precise. However, I find that there remain several matters of clarity and communication, including some issues raised in my previous review. While I still have several concerns, they should be fairly straightforward to address:

1. The question of motion artifact versus real motion state is now addressed in the Discussion, but in many places the word "artifact" is still used to refer to all time-varying changes in FC associated with motion. This is potentially misleading, since as the authors acknowledge, there may be real changes in FC that occur during periods of higher motion. Replacing "motion artifacts" with, say, "motion effects" or "motion impacts" would acknowledge that, as the authors say in the discussion, "motion impact may arise from heterogeneous sources" including artifacts, activation, and FC states.

We had previously used the term "motion artifact" to describe what SHAMAN detects, however the reviewer is correct in pointing out that SHAMAN does not technically distinguish motion artifact from biologically plausible fMRI signal changes related to motion. We have replaced "motion artifact" with "motion," "motion impact," or "motion related signal" throughout the manuscript when describing what SHAMAN detects.

Similarly, the wording "between-participant differences due to head motion" is too strong, as it implies a purely causal relationship between head motion and FC. A better word choice would be "differences associated with head motion" to acknowledge the possibility that head motion may also be indirectly associated with FC due to the influence of various motion-associated traits.

We have replaced "between participant differences **due** to head motion" with "between participant differences **associated with** head motion."

2. In my last review, I requested that the authors present DVARS alongside FD, but the DVARS results are still only presented in the appendix. This is highly inconvenient for readers who wish to consider DVARS alongside FD. The authors should combine Figures 5 and SB.3 (ABCD), as well as Figures SB.2 and SB.4 (HCP). In addition, comparing these two pairs of figures, the “No Censoring” panel should be the same, I assume, across the corresponding figures, but it appears to differ. For example, the left-most group of boxplots in Figure 5 do not appear to match the left-most group of box plots in Figure SB.3. Is there a mistake? (Minor issue: in Figure SB.4 the older terminology of “False Positive” and “False Negative” are used.) In order to facilitate the consideration of DVARS by readers, it is important to present the DVARS results alongside the FD results in the main figures (Fig 5 for ABCD, Fig SB.4 for HCP).

The reviewer makes a good recommendation to place FD and DVARS results in the same figure to facilitate visual comparison. We have updated Figures SB.3 (ABCD) and SB.4 (Human Connectome Project) to show FD (no censoring and censoring) and DVARS (censoring) results side-by-side. Figure SB.4 (below) clearly shows how the principled p-value cutoff method of Afyouni & Nichols (2018) achieves the best reduction in motion overestimation in Human Connectome Project data.

Figure SB.4: Effects of frame censoring on motion impact score in HCP data using FD vs DVARS. Motion impact score (omnibus Stouffer's Z, higher = more motion) for traits of a given category at different levels of motion censoring. The cutoff score for significance at $p < 0.05$ is different for each trait. The range of significance thresholds is indicated by a gray band. **(a)** Motion overestimation score. **(b)** Motion underestimation score.

The “no censoring” panels for FD and DVARS differ because, even without motion censoring, the motion measure (FD or DVARS) is used to regress out between-participant differences associated with head motion. The new, streamlined figures avoid this confusion. The “no censoring” data for DVARS are still available in the supplemental tables.

3. Regarding DVARS, the authors explain that they use the “normalized” version of Afyouni and Nichols (2019), but if I understand correctly, they do not utilize the principled cutoff technique. Afyouni and Nichols proposed two complementary versions of DVARS and a technique for generating the null distribution of each measure, which can then be thresholded at a quantile to identify abnormal volumes with high specificity. One advantage of this approach is that it is invariant to the TR of the data, and hence generalizes across different datasets like the ABCD and HCP. It is important to utilize this improved version of DVARS with the principled cutoff, rather than applying an ad-hoc cutoff.

Figures and tables for DVARS have been updated to use the principled cutoff technique with a motion p-value cutoff of 0.05. The DVARS p-values were calculated using Soroosh Afyouni’s Matlab implementation available on GitHub: <https://github.com/asoroosh/DVARS>

4. In response to my previous feedback, the authors now use a permutation scheme that preserves temporal dependencies. This is vital in order to satisfy the assumption of exchangeability in permutation testing. Unfortunately, results based on “unconstrained permutation”, which violates the assumption of exchangeability, are still presented. There is no guarantee of correct false positive control with a permutation procedure that breaks the underlying structure of the data. Unconstrained permutation is not a valid permutation procedure in this setting, and as such it should be fully replaced by block permutation that preserves temporal autocorrelations.

Data for the unconstrained permutation procedure have now been removed from the manuscript.

5. The title and text of the final results section “Motion Impact Score is Specific to Motion Artifact”, are potentially misleading. The analysis in this section is based on a “fake” non-trait, namely participant ID, which is not related to motion. This analysis does not establish whether Motion Impact Score is “specific to motion artifact” for realistic traits that are themselves associated with motion. Also, it is unclear why $n=100$ participants was considered sufficient for this analysis, while the previous section established that $n = 5,000$ subjects are needed to successfully detect motion overestimation score (and even that many are not sufficient to detect motion underestimation score). My main concern, however, is that the analysis in this section is designed to determine whether Motion Impact Score produces false positives for a null trait, but it cannot determine whether it produces false positives for a more realistic trait. In addition, the wording “specific to motion artifact” implies that SHAMAN can distinguish between motion

artifact and motion-associated signal. It is important to reword both the title and text of this section to accurately describe the conclusions that can be drawn from this particular analysis.

The reviewer highlights one of the key challenges in fMRI head motion research: it is impossible to collect motion-free fMRI data. After considering the reviewer's perspective, we agree that the existing wording is too strong. The revised paragraph (below) makes it clearer that we are testing specificity to association between head motion and a trait, not necessarily **artifact** per se. We also wish to reassure the reviewer that while SHAMAN is specific for motion overestimation in *as few as* 100 participants, we carried the simulations forward to verify that specificity is retained in *as many as* 6,000 participants (see Figure S8).

Motion Impact Score is Specific to Motion-Associated Traits

To be useful, SHAMAN must also be specific, producing not-significant motion impact scores when trait-FC effects **are not associated with motion**. The randomly-assigned participant ID was used as a trait to simulate a trait-FC effect **independent from motion**. Participant ID was not significantly correlated with FD ($r = 0.0006$). We performed bootstrapping with participant ID at different sample sizes to assess SHAMAN's specificity. **SHAMAN did not falsely detect a significant motion impact score at any sample size from 100 to 6,000 participants** (Figure S8).

6. The third section of the Discussion, titled "Motion May Negatively Affect the SNR", includes a claim that is not clearly justified. The authors first observe that motion-FC associations are themselves anti-correlated with FC. Then, they say that "the most parsimonious explanation for this motion-connectivity anti-correlation is that data from participants with higher motion have a lower SNR". I don't see how this follows. Another possible interpretation of the data could be that weaker functional connections (low values of FC) are more susceptible to motion (high values of motion-FC association). If the authors are interested in investigating the relationship between motion and SNR, that could be interrogated more directly. Perhaps the authors's current claim about SNR is correct, but if so, it requires a more clear and convincing explanation.

The reviewer offers compelling alternative interpretations of our findings. An in-depth analysis of SNR is beyond the scope of this manuscript, therefore we have carefully worded the discussion to make it clear that the SNR interpretation is one of many possible interpretations.

While we hypothesize that motion-related signal loss plays a large role in explaining the similarity between the motion-FC effect matrix and average FC matrix, we cannot definitely prove the association from these analyses. Other factors contributing to the observed motion-FC effect matrix may include distance-dependent attenuation of connectivity by motion artifact.⁴⁰ Propensity of an individual to move is itself a stable, heritable trait,⁵⁷⁻⁵⁹ and some⁶⁰ have even proposed that intrinsic differences in FC related to movement are encoded in the FC matrix. While our findings do not exclude the possibility that motion has an intrinsic neurobiological basis, we find it surprising that motion's trait-FC effect would have a larger effect size than any other trait-FC effect. It is more plausible that the motion-FC effect arises from a combination of artifact, signal loss, and biologically-meaningful motion-related traits and FC states.

7. The analyses in this manuscript provide many interesting insights, but most of the conclusions are based on the ABCD. The ABCD has obvious advantages (namely its size) but also limitations due to the specific nature of the study population, data acquisition methods, and data processing/denoising methods. It is not clear how well the conclusions made here generalize. Comparison with the HCP only partly alleviates this limitation, because (1) the results differ quite a lot, comparing Figures 5 and SB.2, and (2) as explained by the authors, there are a number of differences between the datasets, making it difficult to pin down the effects of population, processing, and other factors. Comparing Figures 5 and SB.2, it seems clear that the “no censoring” strategy in the HCP is much less problematic than in the ABCD. The most likely explanation for this is that the HCP uses a fundamentally different denoising strategy (ICA-FIX), which could be more effective at mitigating motion artifact. I do not expect the authors to tease out all of the reasons for the differences between the ABCD and HCP results. However, it is important to (1) acknowledge the lower magnitude and frequency of motion impacts in HCP (the Discuss section “Patterns of Motion Impact Replicate Across Data Sets” currently understates the differences) and (2) acknowledge as a limitation that the main analyses in the paper rely on specific denoising strategies, and that motion impacts and the impacts of frame censoring may be lessened when using ICA-FIX (this is supported by the HCP results) or other advanced denoising strategies like multi-echo ICA.

The reviewer raises some terrific insights about differences between ABCD and other fMRI studies. While we chose to focus on the ABCD study because children are a high-motion population, it is important to demonstrate that our method generalizes to adult studies such as HCP. We include a supplemental discussion that addresses all of the reviewer’s points including (1) the lower magnitude and frequency of motion impacts in HCP and (2) differences in processing techniques, such as the use of ICA-FIX in HCP.

One limitation of our analysis was intrinsic differences in data processing, sample size, sample population, and amount of head motion in ABCD and HCP. The HCP study was processed using ICA-FIX, it had fewer participants, the participants were adults, and the participants had considerably less in-scanner head motion than the children in ABCD, consistent with prior observations of children as a high-motion cohort^{19,22,29}. This made it difficult to disentangle whether lower motion impact scores in HCP were due to superiority of ICA-FIX as a method for mitigating head motion artifact, limitations in SHAMAN’s sensitivity at smaller sample sizes, or lower head motion in the adult cohort.

8. The stated assumption that “the high-motion half of the data will have the same correlation structure as the low-motion half except for artifact from motion” is not “weak” in my opinion, and it is in conflict with other parts of the discussion that describe the diverse potential drivers of motion-FC effects (i.e. “artifact, signal loss, and biologically-meaningful motion-related traits and FC states”). Therefore, it seems necessary to revisit the wording of this assumption.

Following the reviewer's suggestion, we avoid editorializing our choice of assumptions by removing the adjective "weak."

9. In the Frame Censoring section of the Methods, the new FD cutoff of 0.3 is not mentioned, nor is DVARS.

We appreciate the reviewer pointing out this error. The Methods now include the expanded FD cutoff of 0.3.

I believe that this feedback should be straightforward to address, and am hopeful that the authors will find it helpful.